# Dynamic mRNA degradome analyses indicate a role of histone H3K4 trimethylation in association with meiosis-coupled mRNA decay in oocyte aging

Yun-Wen Wu[1,6], Sen Li[2,6], Wei Zheng[3,6], Yan-Chu Li[2,4,6], Lu Chen[1], Yong Zhou[2], Zuo-Qi Deng[1], Ge Lin [3], Heng-Yu Fan [1✉] & Qian-Qian Sha [2,5✉]

A decrease in oocyte developmental potential is a major obstacle for successful pregnancy in women of advanced age. However, the age-related epigenetic modifications associated with dynamic transcriptome changes, particularly meiotic maturation-coupled mRNA clearance, have not been adequately characterized in human oocytes. This study demonstrates a decreased storage of transcripts encoding key factors regulating the maternal mRNA degradome in fully grown oocytes of women of advanced age. A similar defect in meiotic maturation-triggered mRNA clearance is also detected in aged mouse oocytes. Mechanistically, the epigenetic and cytoplasmic aspects of oocyte maturation are synchronized in both the normal development and aging processes. The level of histone H3K4 trimethylation (H3K4me3) is high in fully grown mouse and human oocytes derived from young females but decreased during aging due to the decreased expression of epigenetic factors responsible for H3K4me3 accumulation. Oocyte-specific knockout of the gene encoding CxxC-finger protein 1 (CXXC1), a DNA-binding subunit of SETD1 methyltransferase, causes ooplasm changes associated with accelerated aging and impaired maternal mRNA translation and degradation. These results suggest that a network of CXXC1-maintained H3K4me3, in association with mRNA decay competence, sets a timer for oocyte deterioration and plays a role in oocyte aging in both mouse and human oocytes.

[1] MOE Key Laboratory for Biosystems Homeostasis & Protection and Innovation Center for Cell Signaling Network, Life Sciences Institute, Zhejiang University, 310058 Hangzhou, China. [2] Fertility Preservation Laboratory, Reproductive Medicine Center, Guangdong Second Provincial General Hospital, 510317 Guangzhou, China. [3] Research Center for Reproduction and Genetics in Hunan Province, Reproductive and Genetic Hospital of CITIC-XIANGYA, 410008 Changsha, China. [4] The Second School of Clinical Medicine, Southern Medical University, 510515 Guangzhou, China. [5] School of Basic Medical Sciences, Southern Medical University, 510515 Guangzhou, China. [6] These authors contributed equally: Yun-Wen Wu, Sen Li, Wei Zheng, Yan-Chu Li. ✉email: hyfan@zju.edu.cn; shaqianqian@zju.edu.cn

In mammalian females, the reproductive lifespan is shorter than that of the individuals themselves. This phenomenon is particularly noticeable and significant in humans because of the increased lifespan in civilized societies and becomes dramatically apparent among women seeking assisted reproduction; pregnancy rates significantly decrease in those older than 35 years, mainly as a result of degenerative changes within the aging oocyte, rather than senescent changes in the uterus[1,2].

It has long been recognized that oocyte maturation includes nuclear maturation, including chromosome organization and separation during meiotic cell cycle progression, and cytoplasmic maturation (proper accumulation, localization, compartmentation, and functioning of cytoplasmic organelles, mRNAs, and proteins)[3,4]. It has recently been recognized that the progressive establishment and maintenance of epigenetic marks on DNA and chromatin is an additional aspect of oocyte maturation and is therefore called epigenetic maturation[5,6]. Defects in nuclear and cytoplasmic maturation associated with oocyte aging have been reported. Specifically, cohesin proteins that form ring structures to hold sister chromatids together were loaded on the chromosomes of germ cells during the last round of DNA replication in embryonic gonads. These cohesin rings are prone to loss with aging in oocytes, directly causing precocious chromosome separation during meiotic divisions, and therefore, lead to nuclear maturation defects[7–9]. Compromised spindle assembly checkpoint activity and securin level also cause sister chromatid cohesion loss in aged oocytes[10]. A recent investigation of translational landscape in mouse oocytes demonstrated that decreased translation of a subset of maternal mRNAs is a hallmark of oocyte aging[11]. Specific maternal mRNAs encoding factors involved in meiotic spindle assembly and chromosome segregation were aberrantly translated in oocytes from aged female mice.

Another group of important cytoplasmic mRNAs that are transiently translated during oocyte meiotic maturation are those encoding factors responsible for maternal mRNA removal during the oocyte-to-zygote transition (OZT), including the OZT licensing factor BTG4 and its 3'-UTR adaptor PABPN1L, as well as several catalytic subunits (CNOT6L and CNOT7) of the CCR4-NOT RNA deadenylase[12–17]. The extensive degradation of transcripts mediated by these maternal factors from meiotic resumption to zygotic genome activation is called maternal decay (M-decay), in comparison with the next wave of maternal mRNA clearance that relies on expression of certain early zygotic factors, which is called zygotic decay (Z-decay)[18,19]. Drastic OZT-associated maternal mRNA decay mediated by these factors is intimately associated with the developmental potential of mouse and human preimplantation embryos. Maternal mutations in *Btg4* lead to early zygotic arrest and female infertility in both mice and humans[20]. Furthermore, abnormal accumulation of maternal transcripts caused by the decreased expression of these factors is observed at high frequencies in the development-arrested embryos of human patients seeking assisted reproduction[21]. Despite these findings, the dynamics of mRNA decay in aged human oocytes and the potential involvement of mRNA decay in mammalian oocyte aging remain unclear.

While it is clear that many hallmarks play a crucial role in the progression of aging in the reproductive system, epigenomic changes may be particularly important in oocytes because chromatin states integrate transient and accumulating environmental signals during the long process of oogenesis to influence gene expression and downstream cellular events in oocytes[22]. Transcriptional regulators and chromatin modifiers receive cytoplasmic and extracellular signals and, in turn, alter the responses of the oocytes in an orchestrated manner. In addition, chromatin marks are long-lasting and show a progressive change with age[23]. While these marks can be diluted or erased through divisions in other cell types, they persist in non-dividing oocytes. Thus, they can act as a memory that helps to propagate age-associated oocytic dysfunction. Reduced DNA methylation was observed in oocytes from old female mice, but the majority of DNA methylation changes were not correlated with age-associated transcriptome changes[24]. However, the specific associations between other epigenetic markers and oocyte aging have not been adequately demonstrated.

Methyltransferase or demethylase complexes that modify histone H3K4 trimethylation (H3K4me3) modulate the lifespan of yeast, *C. elegans*, and *Drosophila*[25]. The H3K4me3 global landscape is remodeled during aging in murine hematopoietic stem cells and *Drosophila*[26] and in cellular senescence in human fibroblasts[27]. H3K4me3 domains vary in breadth in a locus- and cell-type-specific manner. Using low-input ChIP–seq approaches, three groups have separately found a non-canonical form of H3K4me3 in MII oocytes and zygotes, which exist as broad peaks at promoters and a large number of distal loci[28–30]. Approximately 22% of the oocyte genome is associated with these broad H3K4me3 domains, which contrast to the typical sharp H3K4me3 peaks restricted to CpG-rich regions of promoters observed in somatic cells. These dynamic changes could provide clues about the context-dependent roles of this marker during oocyte aging, as well as the effect of H3K4me3-modifying enzymes on reproductive lifespan.

There are six histone methyltransferases in mammals that catalyze H3K4 trimethylation: SET domain-containing 1 A/B (SETD1A/B) and mixed lineage leukemia 1–4 (MLL1-4)[31]. The SETD1 complex binds to DNA through its essential subunit CxxC-finger protein 1 (CXXC1)[32,33]. Studies using an oocyte-specific *Cxxc1* knockout mouse strain demonstrated that SETD1-CXXC1 is one of the major methyltransferases that mediates H3K4me3 accumulation in mouse oocytes[34,35]. Deletion of *Cxxc1* in mouse oocytes compromised genomic histone exchange, DNA methylation, gene transcription, and ovulation[36]. Interestingly, the *Cxxc1*-deleted oocytes contained aggregated, brown cytoplasmic granules that resembled those observed in aged oocytes[35]. Despite this observation, the association between CXXC1-mediated H3K4 trimethylation and oocyte aging has not been investigated.

In this study, we determined the meiotic maturation-coupled transcriptome changes in aged human and mouse oocytes and detected deficiencies in transcripts involved in maternal mRNA clearance and histone H3K4 trimethylation. In reinforcing these observations, oocyte-specific deletion of mouse *Cxxc1* caused cellular and transcriptome changes linked to premature aging at a young age. The results suggested that a network of CXXC1-maintained H3K4me3 in association with mRNA decay competence plays a role during aging in both mouse and human oocytes.

## Results

**Transcriptome analyses indicate decreases in maternal mRNA storage in aging human oocytes.** We investigated the intrinsic factors that affect the developmental competence of human oocytes during aging by performing mRNA sequencing analyses in oocytes derived from women of different ages. Gene expression levels were assessed as fragments per kilobase of transcripts per million mapped reads (FPKM), and the relative mRNA copy number was evaluated using the External RNA Controls Consortium (ERCC) spike-in. All samples were analyzed in triplicate, except for two groups of MII oocytes (30–35 and >40, performed in duplicate), due to the scarcity of samples, and showed a high correlation (Rmin = 0.88; Raverage = 0.93; Fig. S1a; Supplementary Data 1). Normalized by ERCC, the results showed that total

mRNA levels were comparable between fully grown GV oocytes from women younger than 30 and 30–35 years old, but continuously decreased thereafter (Fig. 1a).

Compared to oocytes of young women (<30 years old), only 80 and 116 transcripts were increased or decreased by more than 5-fold in the oocytes of women aged 30–35 years (Fig. 1b, left panel). However, with an increase in age, more transcripts were downregulated than upregulated (Fig. 1b, middle and right panels). This trend became even more remarkable when we decreased the thresholds of the analyses to two-fold changes in FPKM, with 86 upregulated and 8264 downregulated transcripts in oocytes of women ≥40 years old, when compared to oocytes of women <30 years old (Fig. 1b). Furthermore, the transcripts decreased by more than 5-fold at different ages and showed significant overlap. With aging, most of the previously downregulated transcripts remained at low levels, and the total number of downregulated transcripts therefore increased due to additional transcripts joining the pool (Fig. 1c).

We further performed gene ontology (GO) analyses of the transcripts that decreased with aging in GV oocytes. Transcripts downregulated in aged oocytes were enriched for translation-related functions, such as structural constituents of ribosomes, translation initiation, rRNA processing, and cytoplasmic translation (Fig. 1d). In addition, transcripts encoding factors involved in cell division and mitochondrial function were also enriched. These results are consistent with previously observed aging-associated defects in human oocytes[11], suggesting that our transcriptome sequencing and analyses are accurate and indicative. Furthermore, transcripts encoding factors of RNA processing, particularly those that regulate mRNA stability, were downregulated in aging oocytes (Fig. 1d), which represents a previously undescribed phenomenon. The age-associated changes in representative genes of these functional categories are shown in a heatmap in Fig. 1e and Fig. S1b.

**Maternal factor-mediated mRNA decay is impaired in human oocytes during aging.** The decrease in transcripts encoding regulators of mRNA stability drew our particular notice as recent studies have indicated that 1) mutations in these factors cause zygotic arrest and female infertility in humans and mice[20]; 2) incomplete maternal mRNA decay during MZT leads to pre-implantation developmental arrest of human embryos even without known gene mutations[21]. Therefore, we hypothesized that the inability of maternal mRNA degradation during oocyte meiotic maturation is a key cytoplasmic factor underlying oocyte aging in humans.

To test this hypothesis, we analyzed age-related transcriptome changes in MII oocytes that underwent large-scale maternal mRNA decay during meiotic maturation. In contrast to the trend observed in GV oocytes (where more and more transcripts were downregulated with age), the number of downregulated transcripts decreased with age in MII oocytes, but an increasing number of transcripts were increased in aged oocytes than in oocytes from young women (<30 years old) (Fig. 2a). Correspondingly, the total mRNA levels based on ERCC-normalized FPKM showed statistically significant increases in MII oocytes from women aged ≥36 years (Fig. 2b). Fully grown oocytes are transcriptionally silent, and aged oocytes generally have lower transcript levels than young oocytes at the GV stage; a rational explanation for the higher transcript levels in aged MII oocytes is that meiosis-coupled mRNA decay is impaired. Consistent with this deduction, the absolute fold change (FC) values of individual transcripts from GV to MII stages also increased in women ≥36 years old, due to lower degrees of mRNA degradation (Fig. 3c). In oocytes from women ≤ 35 years old, 2090 transcripts were significantly degraded during the GV-MII transition (FC[GV/MII] > 5) (Fig. 2d). However, only 223 transcripts were significantly degraded during the GV-MII transition in oocytes from women ≥36 years old (Fig. 2d). We further divided the 2090 M-decay transcripts of young oocytes into two groups: age-sensitive (1909), which failed to be degraded in aged MII oocytes, and age-insensitive (181), where degradation in MII oocytes was not affected by aging. The degradation pattern showed that age-sensitive transcripts in aged oocytes were lower than those in young oocytes at the GV stage, but remained relatively stable during meiotic maturation, and ended up with higher levels than in young oocytes at the MII stage (Fig. 2e, left panel). Age-insensitive transcripts in aged oocytes also had lower levels than in young oocytes at the GV stage, and were degraded during GV-MII transition, albeit to a lesser extent (Fig. 2e, right panel).

**Meiotic maturation-coupled mRNA decay is impaired in oocytes of aged mice.** Next, we investigated whether age-associated maternal mRNA decay defects could also be detected in a mouse model. We collected cumulus cell-enclosed fully grown GV oocytes from 4-week and 14-month-old mice. More than 80% of the oocytes within the cumulus-oocyte complexes had a chromatin configuration of the surrounding nucleolus type, regardless of the mouse age. After 16 h of in vitro culture, 80% oocytes from 4-week-old mice released polar body 1 (PB1) and developed to the MII stage. However, only 40% of the oocytes from 14-month-old mice released PB1. This is consistent with previously published results showing that aged mouse oocytes have low rates of complete meiotic maturation characterized by PB1 emission[37]. Therefore, PB1-absent oocytes could be considered more advanced in the aging process than those that had released PB1.

Next, we performed RNA sequencing on GV and MII oocytes from 4-week and 14-month-old mice, and the oocytes of 14-month-old mice failed to release PB1 after culture (Fig. 3a). Gene expression levels were assessed as FPKM, and the relative mRNA copy number was evaluated using the ERCC spike-in. All samples were analyzed in triplicate, except for the post-culture oocytes from 14-month-old mice without PB1, which were analyzed in duplicate. All triplicates and replicates showed high correlations (Raverage = 0.908; Fig. S2a and Supplementary Data 2). Compared to the transcript levels in oocytes of 4-week mice, the levels of 2430 transcripts decreased, but only 983 transcripts increased 2 fold in GV oocytes of 14-month-old mice (Fig. 3b, left panel). This is consistent with the RNA-seq results in human GV oocytes, where maternal mRNA storage decreased in conjunction with aging. However, in MII oocytes with PB1 release, the trend in age-related mRNA loss was reversed: more transcripts had increased (148) than decreased[12] levels (Fig. 3b, middle panel). The accumulation of maternal transcripts was more significant in PB1-absent post-culture oocytes of 14-month-old mice (Fig. 3b, right panel). We further evaluated global mRNA levels in mouse oocytes based on the ERCC-normalized FPKM values. As expected, the total mRNA level was reduced in oocytes from 4-week-old mice during development from the GV to MII stage. However, mRNA degradation was significantly impaired in oocytes of the 14-month-old mice, with the most dramatic difference observed in oocytes that failed to release PB1 after maturation culture (Fig. 3c). In oocytes of young mice, 5749 transcripts were decreased more than 5 fold during the GV-MII transition (Fig. 3d). Almost all transcripts showed increased levels in aged MII oocytes belonging to these M-decay transcripts (Fig. 3d). Multiple studies in mouse and human oocytes have shown that once mRNA decay is triggered

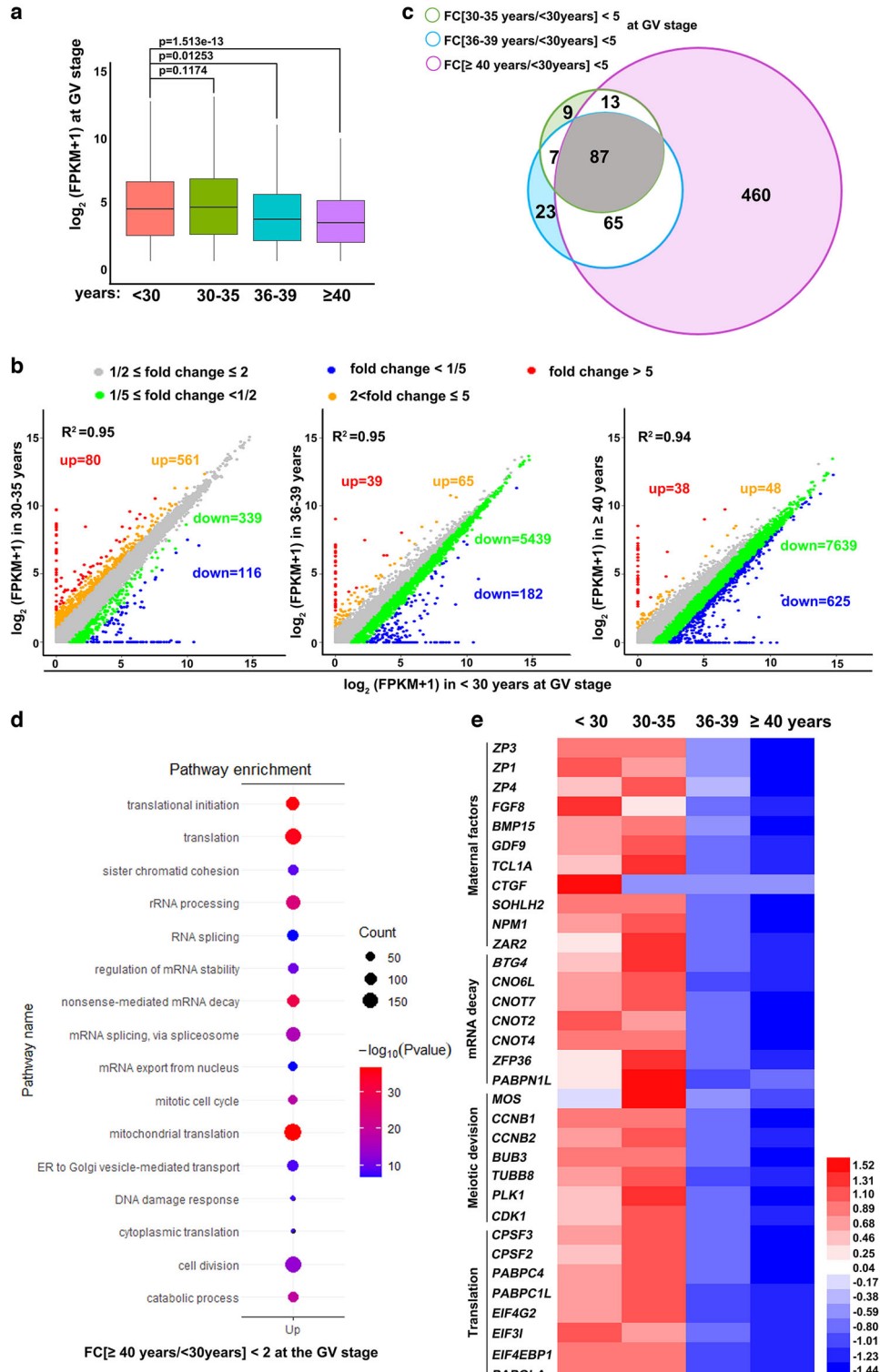

**Fig. 1 Age-associated transcriptome changes in human oocytes at the GV stage. a** Box plot showing the gene expression levels in human GV oocytes of different ages. $n = 3$ biologically independent samples were included in each group. The box indicates the upper and lower quantiles, the thick line in the box indicates the median and whiskers indicates 2.5th and 97.5th percentiles.. *P-values* by a two-tailed Student's *t*-test are indicated. **b** Scatter plots of RNA-seq data illustrating transcriptional changes in human GV oocytes of different ages. Transcripts decreased or increased by more than 2-fold or 5-fold compared with oocytes of women younger than 30 years old were highlighted with different colors. **c** Venn diagram showing the overlap of transcripts that decreased more than 5 fold in GV oocytes from women of different ages, compared with oocytes of women younger than 30 years old. **d** Gene ontology analysis of transcripts significantly downregulated in aged GV oocytes. *P-values* by a two-tailed Student's *t*-test are indicated. **e** Heat maps showing the level changes in representative transcripts in the indicated functional categories.

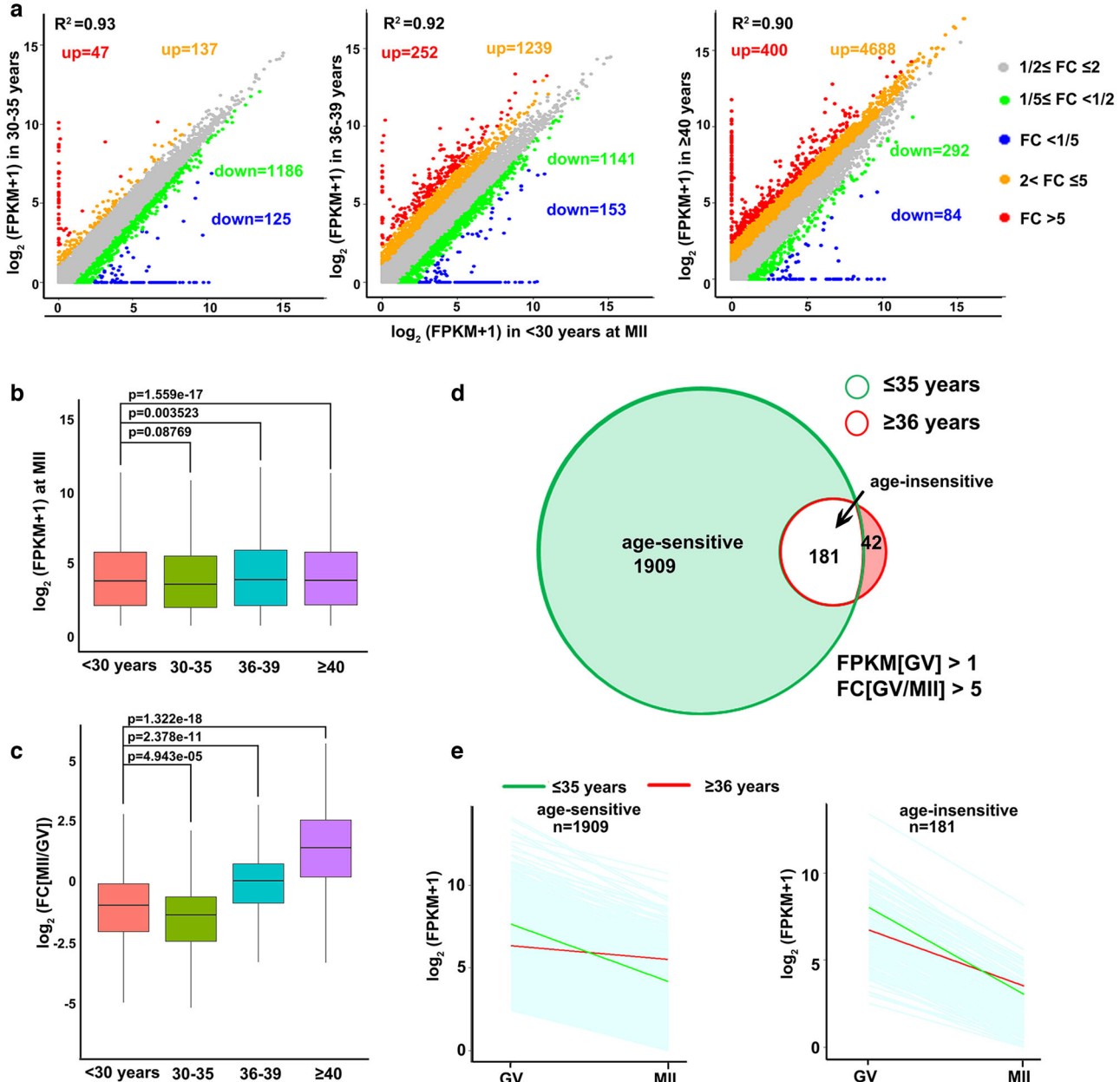

**Fig. 2 Age-associated changes in mRNA degradation during GV-MII transition in human oocytes. a** Scatter plots of RNA-seq data illustrating the transcriptional changes in human MII oocytes of different ages. Transcripts that decreased or increased by more than 2-fold or 5-fold compared with oocytes of women younger than 30 years old are highlighted with different colors. **b** Box plot showing gene expression levels in human MII oocytes of different ages. $n = 3$ biologically independent samples were included in each group. The box indicates the upper and lower quantiles, the thick line in the box indicates the median and whiskers indicates 2.5th and 97.5th percentiles.. *P-values* by a two-tailed Student's *t*-test are indicated. **c** Box plot showing fold changes in mRNA levels in MII versus GV oocytes from women of different ages. $n = 3$ biologically independent samples were included in each group. The box indicates the upper and lower quantiles, the thick line in the box indicates the median and whiskers indicates 2.5th and 97.5th percentiles. *P-values* by a two-tailed Student's *t*-test are indicated. **d** Venn diagram showing the overlap of M-decay transcripts (decreased 5 fold from GV to MII stage) in oocytes from women younger and older than 35 years. **e** Degradation patterns of human maternal transcripts during the GV-MII transition in oocytes derived from women aged ≤ and >35 years. Each light blue line represents the expression levels of one gene, and the middle red and green lines represent the median expression levels of the group. Transcripts with FPKM >1 at the GV stage were selected and analyzed.

after GVBD, the clearance of transcripts is unaffected by spindle organization and PB1 release[21,38]. Therefore, the severe defect in maternal mRNA clearance in PB1-absent oocytes is highly likely to be associated with a more advanced aging process.

Compared to the M-decay transcripts in oocytes of young mice, fewer transcripts were efficiently degraded in aging oocytes during

the GV-MII transition, with or without PB1 (Fig. 3e). Therefore, as in human oocytes, there were age-sensitive (1759) and age-insensitive (1555) M-decay transcripts in mice. Age-sensitive transcripts were removed in young oocytes during GV-MII transition, but remained relatively stable in aged oocytes (Fig. 3f, left panel), whereas age-insensitive transcripts were degraded in both young and aged oocytes during meiotic maturation, albeit to a

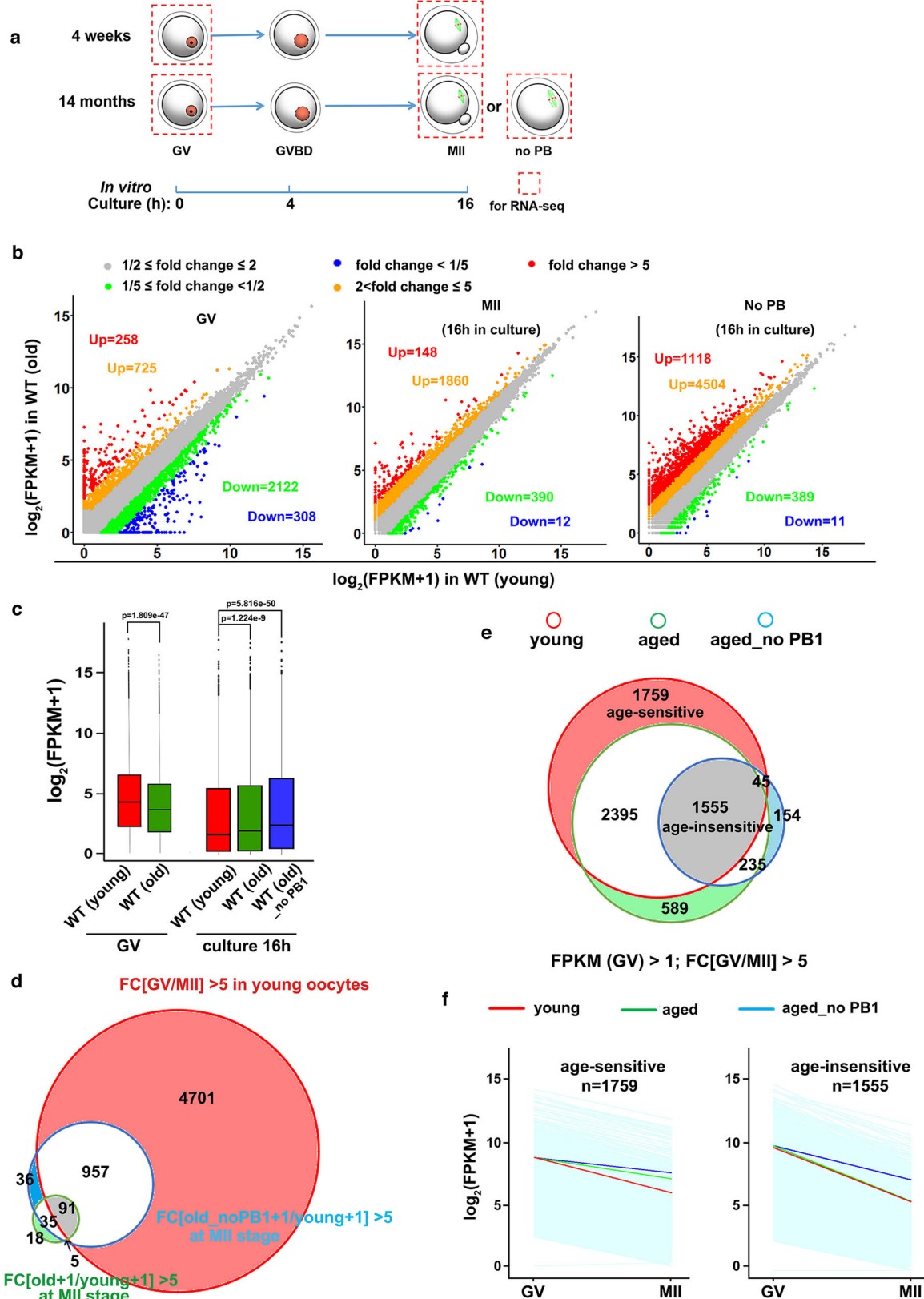

lesser extent in aged oocytes that failed to release PB1 (Fig. 3f, right panel). However, in contrast to observations in human oocytes, in aged mouse oocytes, these M-decay transcripts did not show lower expression levels at the GV stage compared to young oocytes (Fig. 3c, f). This issue will be revisited in the Discussion section.

**Potential involvement of histone H3 lysine-4 trimethylation in oocyte aging**. In the following analyses, we investigated the potential mechanisms that cause aging-associated transcriptome changes in oocytes, i.e. downregulation of maternal mRNA transcription at the GV stage and impaired maternal mRNA decay at

**Fig. 3 Meiotic maturation-coupled mRNA decay in oocytes of young and aged mice. a** A diagram showing mouse oocyte samples collected for RNA-seq analysis **b** Scatter plots of RNA-seq data illustrate the transcriptional changes in oocytes of 4-week-and 14-month mice before (fully grown germinal vesicle (GV) stage) and after in vitro maturation culture (with or without polar body 1 (PB1)). Transcripts decreased or increased more than 5-fold compared with WT (4-week) are highlighted in blue or red, respectively. **c** $n = 3$ biologically independent samples were included in young and old WT GV group and old WT MII (with PB1) group. $n = 2$ biologically independent samples were included in WT old MII (without PB1) group. The box indicates the upper and lower quantiles, the thick line in the box indicates the median, and the whiskers represent 2.5th and 97.5th percentiles. The dots outside the whiskers of the box plot represent the outliers. *P-values* by a two-tailed Student's *t*-test are indicated. **d** Venn diagram showing the overlap of transcripts that decreased from GV to MII stage in oocytes of WT (4-week) mice and transcripts accumulated in oocytes (with or without PB1) in 14-month mice after a 16 h maturation culture. **e** Venn diagram showing the overlap of transcripts that decreased 5 fold from GV to MII stage in oocytes of 4-week and 14-month WT mice. Transcripts with FPKM > 1 at the GV stage were selected and analyzed. **f** Degradation patterns of mouse maternal transcripts during the GV-MII transition in oocytes derived from young and aged mice. Each light blue line represents the expression levels of one gene, and the middle red, green, and blue lines represent the median expression levels of the group.

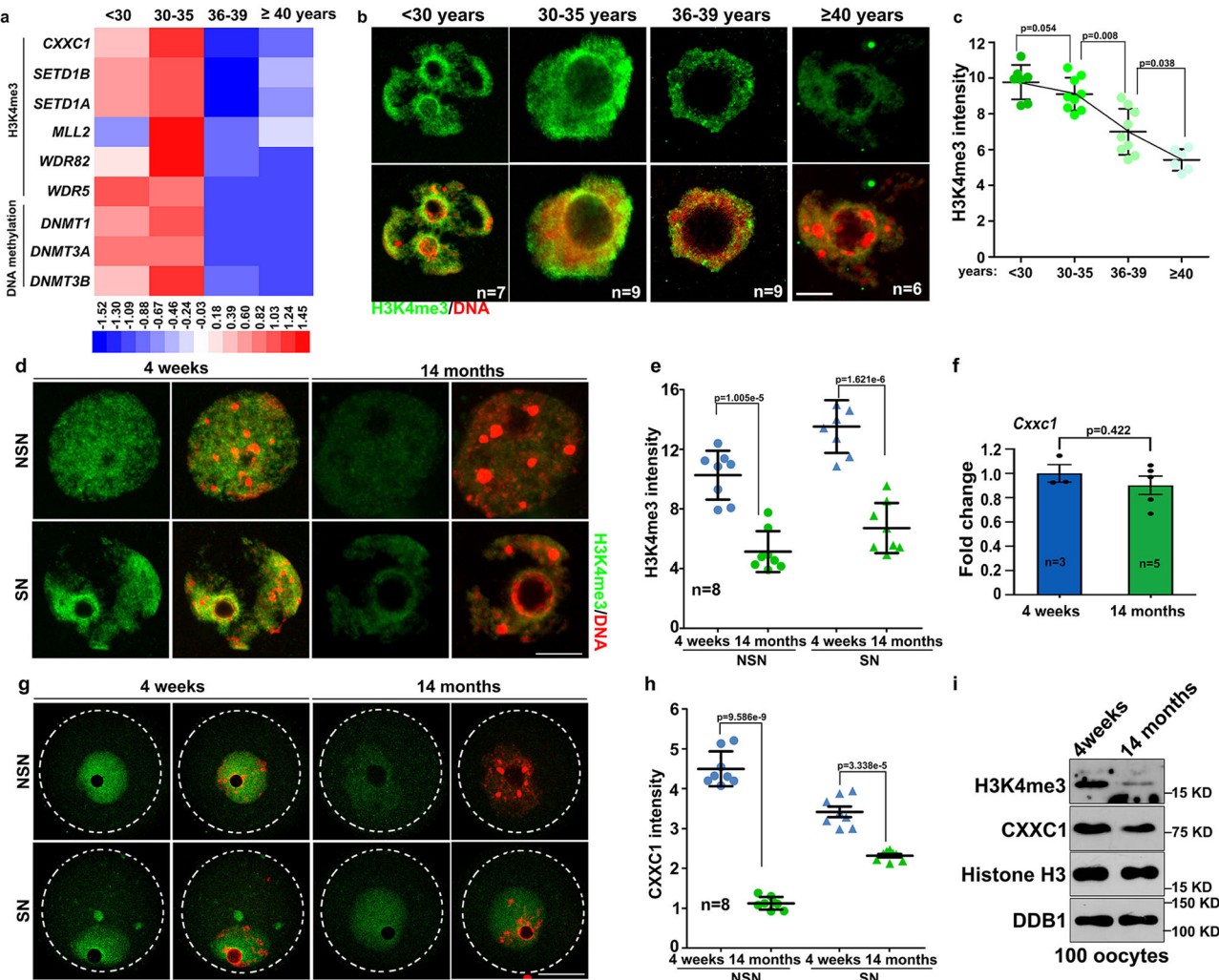

**Fig. 4 Age-associated changes of trimethylated histone H3 lysine-4 (H3K4me3) levels in human and mouse oocytes. a** Heat maps showing level changes in transcripts encoding epigenetic regulators in the indicated functional categories. **b** Immunofluorescence of H3K4me3 in GV stage-arrested oocytes from women at the indicated ages. The number of examined oocytes is indicated (n). Only the nuclei of oocytes are displayed. Scale bar = 10 μm. **c** Quantification of H3K4me3 immunofluorescence signals in (**b**). **d** Immunofluorescence of H3K4me3 in GV stage-arrested fully grown oocytes derived from 4-week and 14-month mice. Only the nuclei of oocytes are displayed. Scale bar = 10 μm. NSN: oocyte with non-surrounded nucleolus; SN: oocyte with surrounded nucleolus. **e** Quantification of H3K4me3 immunofluorescence signals (**d**). $n = 8$ oocytes for each experimental group. **f** Quantitative RT-PCR results showing *Cxxc1* mRNA levels in GV stage-arrested fully grown oocytes derived from 4-week and 14-month-old mice. $n = 3$ biologically independent samples in oocytes of 4 weeks old female mice; $n = 5$ biologically independent samples in oocytes of 14 months old mice. **g** Immunofluorescence of CXXC1 in GV stage-arrested fully grown oocytes derived from 4-week and 14-month mice. Oocytes are outlined by broken circles. Scale bar = 20 μm. **h** Quantification of CXXC1 immunofluorescence signals (**g**). $n = 8$ oocytes for each experimental group. **i** Western blots of indicated proteins in fully grown GV oocytes derived from 4-week and 14-month mice. Histones H3 and DDB1 were used as the loading controls. In **c, e, f** and **h**, data are presented as mean values ± SEM *P* by two-tailed Student's *t*-test. n.s.: non-significant.

the MII stage. H3K4me3 is a key epigenetic modification that regulates transcriptome changes and determines oocyte developmental potential[5]. However, the RNA-seq results indicated that transcripts encoding subunits of the COMPASS complexes, including CXXC1, SETD1A, SETD1B, and MLL2, were downregulated in human oocytes of advanced age (Fig. 4a). Consistent with these results, we observed significant decreases in H3K4me3 levels in oocytes from women aged ≥36 years (Fig. 4b, c). The curve of H3K4me3 decrease (Fig. 4c) was like the curve of age-associated decrease in pregnancy rates in assisted reproduction[39].

H3K4me3 levels in the nucleus also significantly decreased in aged mouse oocytes derived from 14-month-old mice (Fig. 4d, e). Although its mRNA level did not change with aging (Fig. 4f), the expression of CXXC1 protein, which is essential for H3K4me3 accumulation during oogenesis, was downregulated in older mouse oocytes (Fig. 4g–i). Western blot results also indicated lower H3K4me3 and CXXC1 levels in oocytes from aged mice than in oocytes from young mice (Fig. 4i). Unfortunately, the commercially available CXXC1 antibody failed to detect endogenous CXXC1 in human oocytes (data not shown).

Collectively, these results suggest a potential association between CXXC1-mediated H3K4 trimethylation and oocyte aging in both mice and humans.

**_Cxxc1_-deletion in oocytes leads to cytoplasmic defects resembling those in aged oocytes.** In the following experiments, we employed an oocyte-specific _Cxxc1_ knockout mouse model to investigate the potential role of H3K4me3 in preventing oocyte aging, because 1) CXXC1 is a DNA-binding subunit of the SETD1A/B histone H3K4 methyltransferase and is required to maintain normal H3K4me3 levels in oocytes, and 2) _Cxxc1_ mRNA and protein levels were downregulated in aged mouse and human oocytes. The _Cxxc1_ gene was deleted in postnatal oocytes by crossing the previously reported _Cxxc1_ floxed mice with _Gdf9-Cre_ mice. Successful removal of CXXC1 and reduction of H3K4me3 in the oocytes of the resultant _Cxxc1_$^{fl/fl}$;_Gdf9-Cre_ mice (short as _Cxxc1_$^{oo-/-}$) had been previously confirmed[34,36].

Oocyte aging is characterized by mitochondrial dysfunction and increased reactive oxygen species (ROS) in the ooplasm[37]. Compared with oocytes from 4-week-old mice, oocytes from 14-month-old mice showed increased ROS levels and low mitochondrial membrane potential. Similar cytoplasmic phenotypes were also observed in oocytes from 1-month-old _Cxxc1_$^{fl/fl}$;_Gdf9-Cre_ mice (Fig. 5a–d). In growing oocytes with a non-surrounded nucleolus chromatin configuration, the general transcription activity decreased with aging, as detected by an EU incorporation assay (Fig. 5e, f). Active histone H3.3 exchange in oocyte chromatin is an indicator of chromatin tightness and accessibility. We exogenously expressed HA-tagged histone H3.3 by mRNA microinjection in oocytes. Significantly more de novo synthesized H3.3 was incorporated into the chromatin of young oocytes than in old oocytes (Fig. 5g, h).

**_Cxxc1_-deletion in oocytes leads to defects in mRNA decay resembling those in aged oocytes.** To understand the impact of _Cxxc1_ deletion on the oocyte transcriptome during meiotic maturation, we subjected WT and _Cxxc1_ null oocytes to global RNA-seq analyses. Gene expression levels were assessed as FPKM, and the relative mRNA copy number was evaluated using the ERCC spike-in. All samples were analyzed in triplicate and showed a high correlation (R$_{average}$ = 0.899; Fig. S2a; Supplementary Data 3). Compared to WT, 1000 and 1543 transcripts were increased or decreased more than 2-fold in _Cxxc1_ null oocytes at the GV stage, respectively (Fig. 6a). In contrast, after in vitro maturation culture for 16 h, more transcripts showed

higher (4380) than lower (1202) levels compared to the WT in _Cxxc1_ null oocytes. The total mRNA levels decreased in _Cxxc1_ null oocytes at the GV stage, but were higher than those in WT oocytes at the MII stage, apparently due to insufficient degradation (Fig. 6b). The trend was similar to, but more significant than that in WT oocytes of advanced age. In gene set enrichment analyses, the transcripts downregulated in _Cxxc1_ null GV oocytes had substantial overlap with those downregulated in aged WT oocytes at the GV stage (Fig. 6c). Of the 5749 M-decay transcripts identified in young WT oocytes, 4256 were significantly accumulated in _Cxxc1_ null oocytes, and only 1498 were sufficiently degraded during meiotic maturation (Fig. 6d, e). The degradation patterns of _Cxxc1_-sensitive and _Cxxc1_-insensitive M-decay transcripts are shown in Fig. 6f; the transcripts accumulated in _Cxxc1_ null and aged WT oocytes after meiotic maturation had more advanced overlaps (Fig. 6g). RT-qPCR results confirmed the RNA-seq data, and indicated that the clearance of previously reported M-decay transcripts during meiotic maturation was compromised after aging or _Cxxc1_ knockout (Fig. S2b).

**CXXC1 maintains cytoplasmic translation of maternal factors required for mRNA decay.** Although M-decay was impaired in aged mouse oocytes, the mRNA levels of BTG4, CNOT7, and CNOT6L did not decrease with advanced age, as detected by RNA-seq and RT-qPCR (Fig. 7a; Fig. S2c). Previous studies have shown that the expression of these MZT factors is regulated by meiotic cell cycle-coupled translational activation of pre-existing transcripts[3]. The results of the HPG incorporation assay indicated that the general protein translation activity also decreased with aging in maturing mouse oocytes (Fig. 7b, c).

To specifically detect the translational activation of specific transcripts encoding RNA deadenylation-related factors, we cloned the 3′-UTR of mouse _Btg4_ (3′-UTR$_{mBtg4}$) and inserted it into a _Flag-Gfp_ tagged eukaryotic expression vector. Next, we transcribed unpolyadenylated mRNAs encoding _Flag-Gfp_—3′-UTR$_{mBtg4}$ and microinjected the mRNAs into GV oocytes; in vitro-polyadenylated mRNA encoding mCherry cDNA was co-injected as a positive control. Microinjected oocytes were further cultured for 14 h in M16 medium with or without milrinone, which represses GVBD (Fig. 7d). GFP fluorescence and FLAG western blot results indicated that the expression of the reporter protein driven by _Btg4_ 3′-UTR was compromised in oocytes from aged mice (Fig. 7e–g). In addition to BTG4, we also performed 3′-UTR reporter assays for other maternal transcripts encoding MZT-related factors, including _Cnot6l_ (3′-UTR$_{mCnot6l}$), _Pabpn1l_ (3′-UTR$_{mPabpn1l}$), and _Cnot7_ (3′-UTR$_{mCnot7}$), in oocytes isolated from 4-week or 14-month-old mice. Both florescence (Fig. 8a, b, d, e, g, h) and western blot (Fig. 8c, f, and i) results showed that the meiotic maturation-associated translational activation of these transcripts was compromised in oocytes from mice with advanced age. Polysome isolation and sequencing (Del Llano, E., et al., Aging Cell 2020) results also showed that translation of mRNAs encoding BTG4 and CCR4-NOT subunits decreased in aged mouse oocytes. Although the standard errors between different samples was too large to make significant difference, these results suggested decreased translation of these transcripts in aged oocytes (Fig. S2d).

Reporter assays were performed in _Cxxc1_ null oocytes to detect the translation of these maternal factors required for mRNA decay, including CNOT6L (Fig. 9a, b), BTG4 (Fig. 9c, d), PABPN1L (Fig. 9e, f), and CNOT7 (Fig. 9g, h). Similar to the results observed in aged WT oocytes, quantification results of GFP signals driven by 3′-UTR reporters indicated that translational activation of these maternal transcripts was weakened in oocytes isolated from 4-week-old _Cxxc1_$^{fl/fl}$;_Gdf9-Cre_ mice.

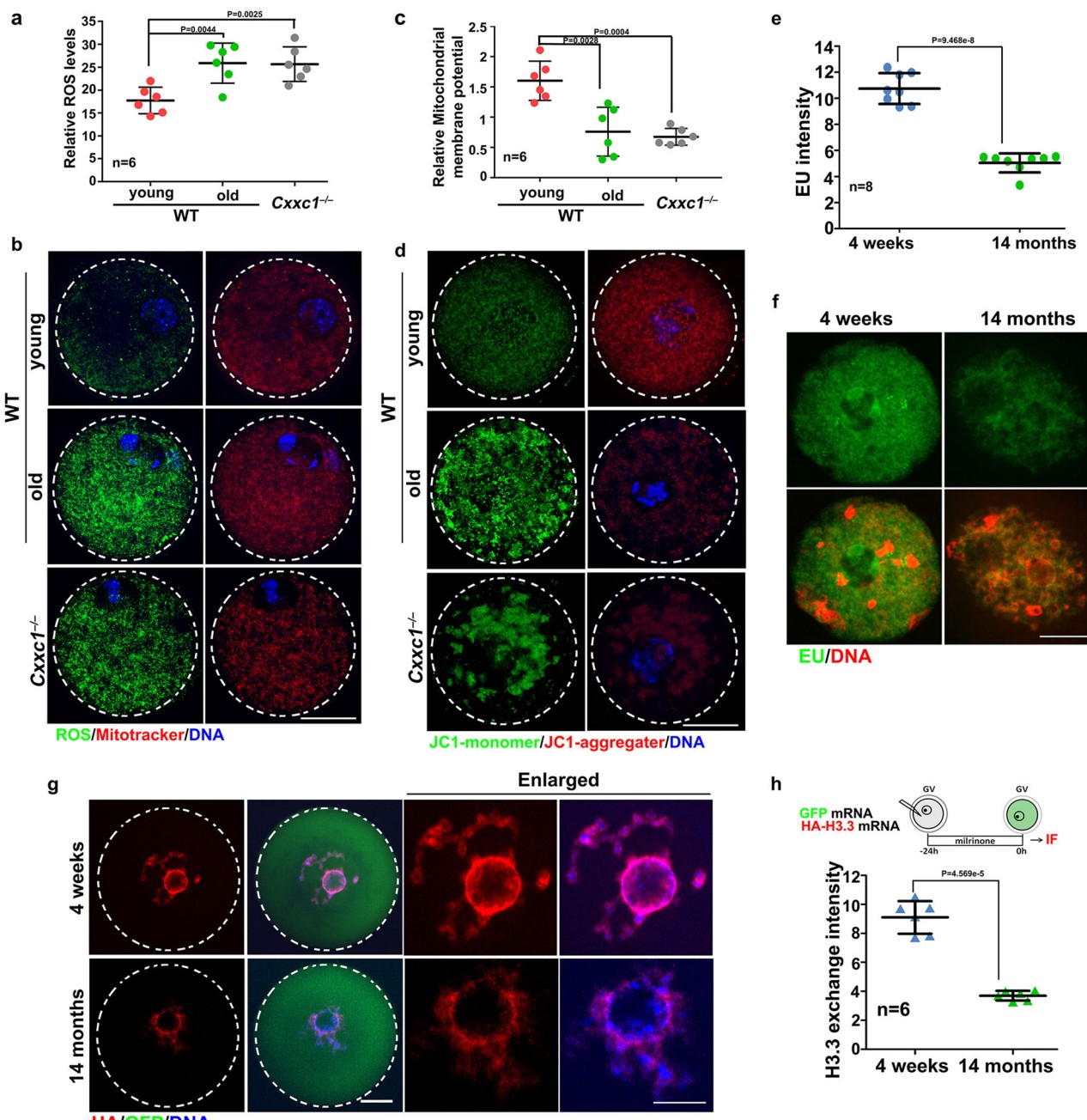

**Fig. 5 Aging-associated cytoplasmic changes in WT and *Cxxc1*-deleted oocytes. a** Detection of reactive oxygen species (ROS) in WT and *Cxxc1*-deleted oocytes after in vitro maturation culture. Mitochondria were stained with MitoTracker. Scale bar = 20 μm for all panels. **b** Quantification of ROS signals in (**a**). *n* = 6 oocytes per experimental group. **c** JC1 staining in WT and *Cxxc1*-deleted oocytes. Scale bar = 20 μm for all panels. **d** Mitochondrial membrane potential (MMP) in oocytes isolated from mice of the indicated genotypes, according to JC-1 staining in (**c**). *n* = 6 oocytes per experimental group. **e** Detection of newly synthesized RNA by 5-ethynyl uridine (EU) incorporation in growing oocytes with non-surrounded nucleolus derived from 4-week and 14-month mice. Only the nuclei of oocytes are displayed. Scale bar = 10 μm for all panels. **f** Quantification of EU signals (**e**). *n* = 8 oocytes for each experimental group. **g** Oocytes were subjected to mRNA microinjection of HA-tagged histone H3.3. An mRNA encoding GFP was co-injected as a positive control for microinjection and expression. The incorporation of histone variants was visualized using anti-HA antibody staining. Scale bar = 10 μm for all panels. **h** Quantification of chromatin-incorporated HA-histone H3.3 signals in (**g**). *n* = 6 oocytes for each experimental group. In **a, b**, **e**, and **h**, data are presented as mean values ± SEM. *P* by two-tailed Student's *t*-test.

Maternal mRNA translation is highly dependent on CPSF. *Cpsf4* and *Cpsf4l* are highly expressed in oocytes. We analyzed the mRNA expression levels of *Cpsf4* and *Cpsf4l* using RT-qPCR, and protein level of CPSF4 by immunofluorescence, respectively. The results showed that the *Cpsf4* mRNA level remains unchanged among WT young, WT old, and *Cxxc1* null oocytes, but *Cpsf4l* mRNA level is significantly decreased in WT old and *Cxxc1* null oocytes (Fig. S3a, b). Meanwhile, the immunofluorescence showed that the protein level of CPSF4 is significantly decreased in WT old and *Cxxc1* null oocytes at GV and MII stage (Fig. S3c-f).

Taken together, these results suggest that the weakened translation of preexisting mRNAs, particularly those encoding

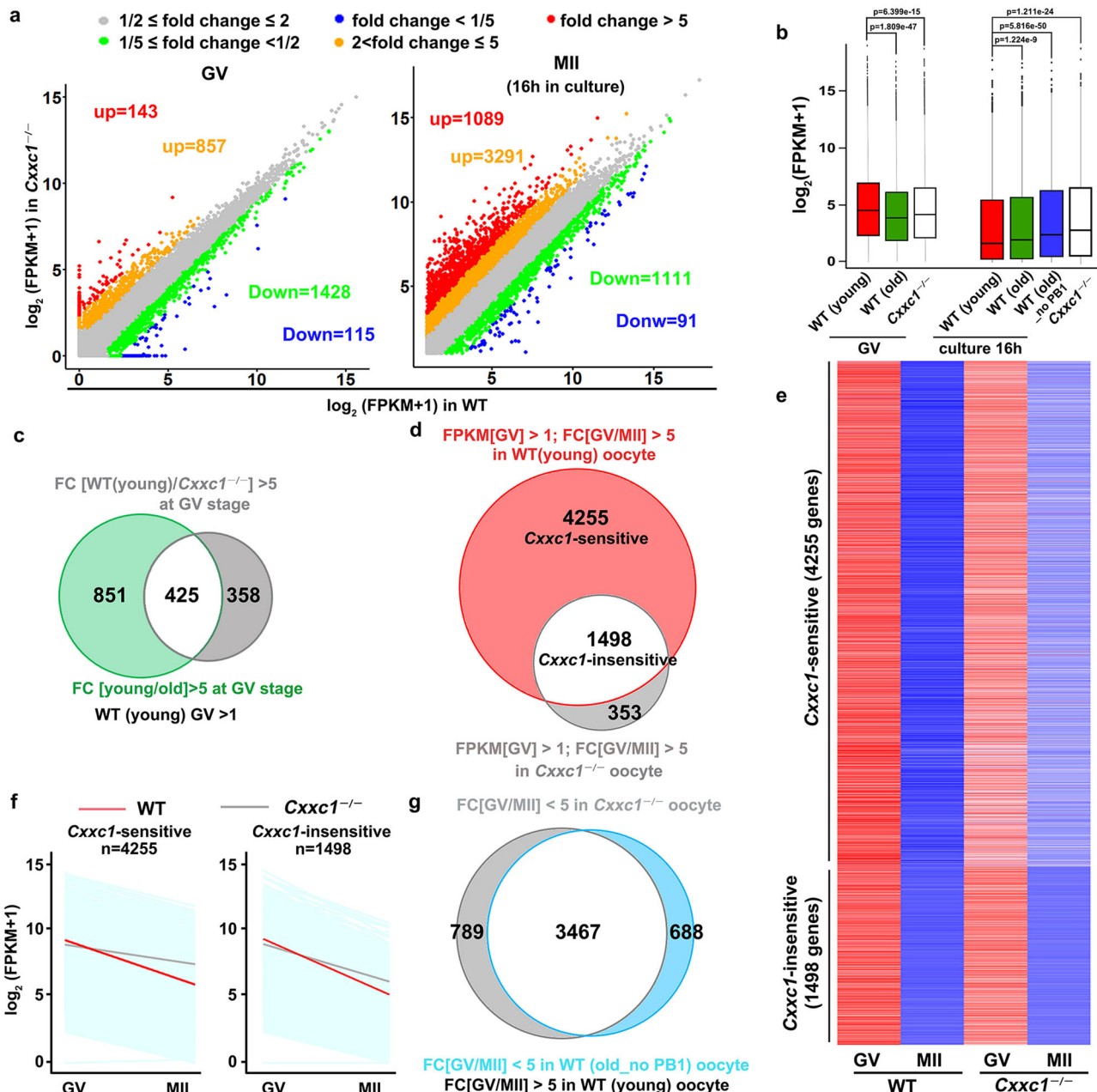

**Fig. 6 Meiotic maturation-coupled mRNA decay in *Cxxc1*-deleted oocytes. a** Scatter plots of RNA-seq data illustrating transcriptional changes in *Cxxc1*-deleted oocytes in the GV and metaphase II (MII) stages. **b** Box plot showing gene expression levels of WT (4-week and 14-month) and *Cxxc1*-deleted (4-week) oocytes before and after in vitro maturation culture. $N = 3$ biologically independent samples were included in young and old WT GV group, *Cxxc1*-deleted GV group, old WT MII (with PB1) group and *Cxxc1*-deleted MII group. $n = 2$ biologically independent samples were included in WT old MII (without PB1) group. The box indicates the upper and lower quantiles, the thick line in the box indicates the median, and the whiskers represent the 2.5th and 97.5th percentiles. The dots outside the whiskers of the box plot represent the outliers. *P-values* by a two-tailed Student's *t*-test are indicated. **c** Venn diagram showing the overlap of transcripts that decreased more than 5 fold in WT (14-months) and *Cxxc1*-deleted oocytes (4-week) compared with WT (4-week) oocytes at the GV stage. **d** Venn diagram showing the overlap of transcripts that decreased 5 fold from the GV to MII stage in 4 week WT and *Cxxc1*-deleted oocytes. Transcripts with FPKM >1 at the GV stage were selected and analyzed. **e** Heat maps showing level changes in maternal transcripts removed during oocyte meiotic maturation in a *Cxxc1*-dependent and independent manner. **f** Degradation patterns of mouse maternal transcripts during GV-MII transition in oocytes derived from 4-week-old WT and *Cxxc1^oo-/-^* mice. Each light blue line represents the expression level of one gene, and the middle red and gray lines represent the median expression levels of the cluster. **g** Venn diagram showing the overlap of M-decay transcripts that were stabilized during in vitro maturation in 14 month WT (no PB) and 4-week *Cxxc1*-deleted oocytes.

factors involved in mRNA deadenylation, is the major cause of impaired maternal mRNA decay in maturing mouse oocytes of advanced age or with *Cxxc1* deletion.

## Discussion

A decrease in oocyte developmental potential is a major obstacle for successful pregnancy in women of advanced age. While the

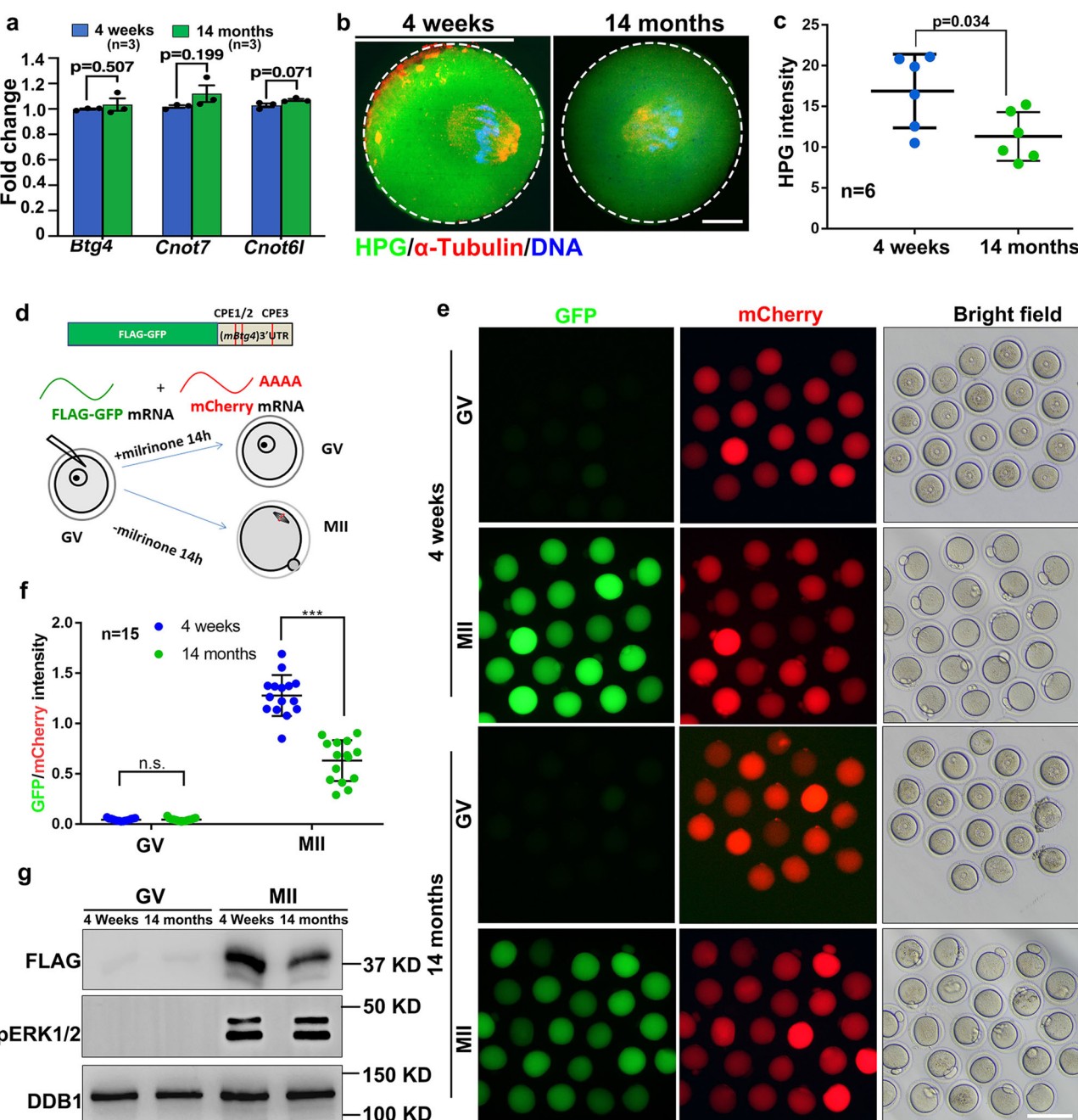

**Fig. 7 Age-associated changes of mRNA translation activities in mouse oocytes. a** Quantitative RT–PCR results showing the relative levels of the indicated transcripts in GV oocytes derived from 4-week and 14-month mice. $n = 3$ biological replicates. data are presented as mean values ± SEM. Statistical analysis was performed using two-tailed Student's $t$-test. n.s.: non-significant. **b** HPG fluorescent staining showing the protein synthesis activity in oocytes derived from 4-week and 14-month mice. Scale bar = 10 μm. **c** Quantification of HPG signals in (**b**). $n = 6$ oocytes for each experimental group. Data are presented as mean values ± SEM. $P$ by two-tailed Student's $t$-test. **d**, **f**: Illustration (**d**), fluorescence microscopy (**e**), quantification of the fluorescent signals (**f**), and western blot (**g**) results showing the translation activities of the *Btg4* 3'-UTR in GV-arrested (maintained by milrinone) or MII-arrested (released from milrinone) oocytes. The GFP signal indicated translational activation of the *Btg4* 3'-UTR. An in vitro transcribed and polyadenylated mCherry mRNA was co-injected as a positive control. Scale bar: 100 μm (**e**). $n = 15$ oocytes for each experimental group. Data are presented as mean values ± SEM. $P$ by two-tailed Student's $t$-test (**f**).

age-associated transcriptome changes in mouse and human oocytes have been reported in several previous studies[11,24,39–41], we approached the long-lasting question of age-related oocyte quality decrease from two new and interacting perspectives; namely, genomic histone methylation changes and meiotic maturation-coupled mRNA degradome.

Previous studies analyzed mouse and human oocyte transcriptomes in correlation with advanced maternal age, mainly focusing on the two meiotic arrest points, GV and/or MII stages[40–42]. Recent investigations have demonstrated that oocytes undergo meiotic maturation, fertilization, and genome reprogramming in the absence of de novo transcription but are accompanied by drastic maternal mRNA degradation[3,43,44]. Efficient maternal mRNA clearance correlates with successful zygotic development in both mice and humans[18,21]. As such, conventional snapshot transcriptome analyses, although

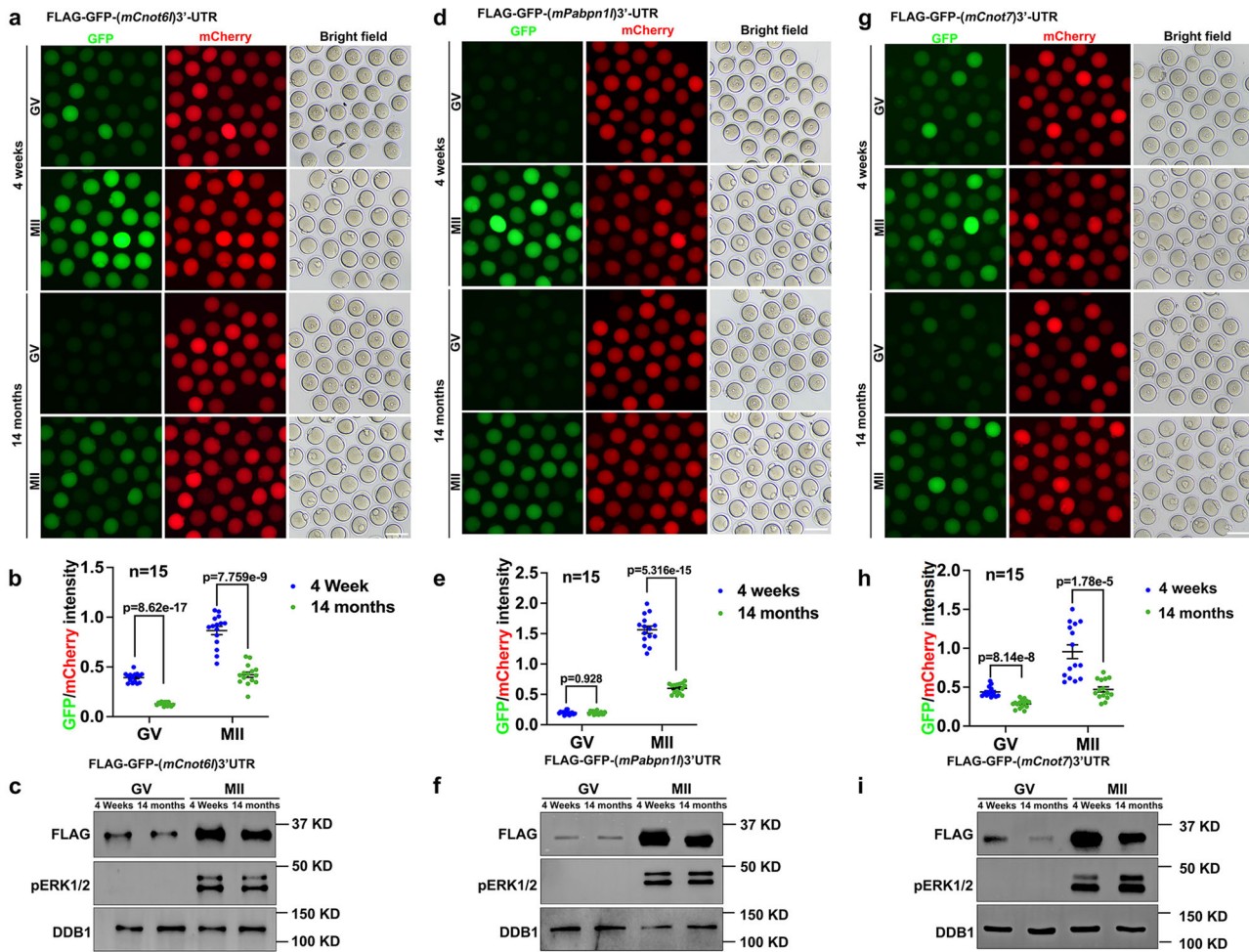

**Fig. 8 Translational activities of transcripts encoding factors related to maternal mRNA clearance in oocytes of 4-week or 14-month-old WT mice.**
Fluorescence microscopy (**a**), quantification of the fluorescent signals (**b**), and western blot (**c**) results showing the translation activities of the *Cnot6l* 3′-UTR in GV-arrested or MII-arrested oocytes of 4-week or 14-month-old mice. The GFP signal indicated translational activation of the *Btg4* 3′-UTR. An in vitro transcribed and polyadenylated mCherry mRNA was co-injected as a positive control. Fluorescence microscopy (**d**), quantification of the fluorescent signals (**e**), and western blot (**f**) results showing the translation activities of the *Pabpn1l* 3′-UTR in GV or MII oocytes of 4-week or 14-month-old mice. Fluorescence microscopy (**g**), quantification of the fluorescent signals (**h**), and western blot (**i**) results showing the translation activities of the *Cnot7* 3′-UTR in GV or MII oocytes of 4-week or 14-month-old mice. Scale bar = 100 μm (**a, d, g**). In **b, e** and **h**, data are presented as mean values ± SEM. *P* by two-tailed Student's *t*-test. *n* = 15 oocytes for each experimental group.

undoubtedly valuable at the time, might not reflect the dynamic transcriptome changes, or the mRNA degradome, when comparing young and aged oocytes. To our knowledge, this is the first study to purposely interrogate the whole mRNA degradome in human oocytes of different maternal ages. A pioneering study revealed that the differentially expressed transcripts between oocytes derived from young and old mice significantly increased from the GV (~5%) to MII (~33%) stages, suggesting that the meiotic maturation-coupled maternal mRNA degradome is dramatically altered during aging[40]. Our current results extended this observation from mice to humans, further addressed the molecular bases causing the age-associated alteration of maternal mRNA degradome, and emphasized the biological consequence of insufficient maternal mRNA decay (Fig. 10 and Fig. S4).

Transcriptome and translatome changes related to spindle assembly and chromosome separation defects in aged oocytes have been described in prior studies[11,42]. However, the mechanism underlying these abnormalities is unclear. In this study, we suggest that histone H3K4me3, a widespread and multifunctional epigenetic marker, is likely to be an important cue that links temporal and environmental inputs to age-

associated cytoplasmic changes. A dramatic decrease in histone H3K4me3 levels, as well as reduced expression of its upstream regulators, were detected in oocytes of aged mice (14 months old) and humans (more than 35 years old). Notably, among women seeking assisted reproduction, pregnancy rate also significantly decreased in those older than 35 years, suggesting that the oocytic H3K4me3 level and maternal mRNA removal capacity are important factors involved in oocyte aging. In support of this hypothesis, oocyte-specific deletion of *Cxxc1*, which encodes a key factor of the SETD1 histone H3 methyltransferase complex, led to a wide spectrum of premature aging-related phenotypes, including 1) aneuploidy after meiotic division, anovulation, and, ultimately, female infertility;[34–36] 2) some cytoplasmic changes, including increases of ROS levels, aggregation of mitochondria, decreases of mitochondrial DNA copy numbers as well as mitochondrial membrane potential, were observed in both aged oocytes and *Cxxc1*-null oocytes (Fig. 5). 3) At the molecular level, meiotic maturation-associated mRNA decay was impaired in both aged oocytes (Fig. 3) and *Cxxc1*-null oocytes (Fig. 6). This is due to the inefficient translation of transcripts encoding mRNA turnover factors (*Btg4*, *Pabpn1l*, *Cnot6l*) in both aged oocytes and

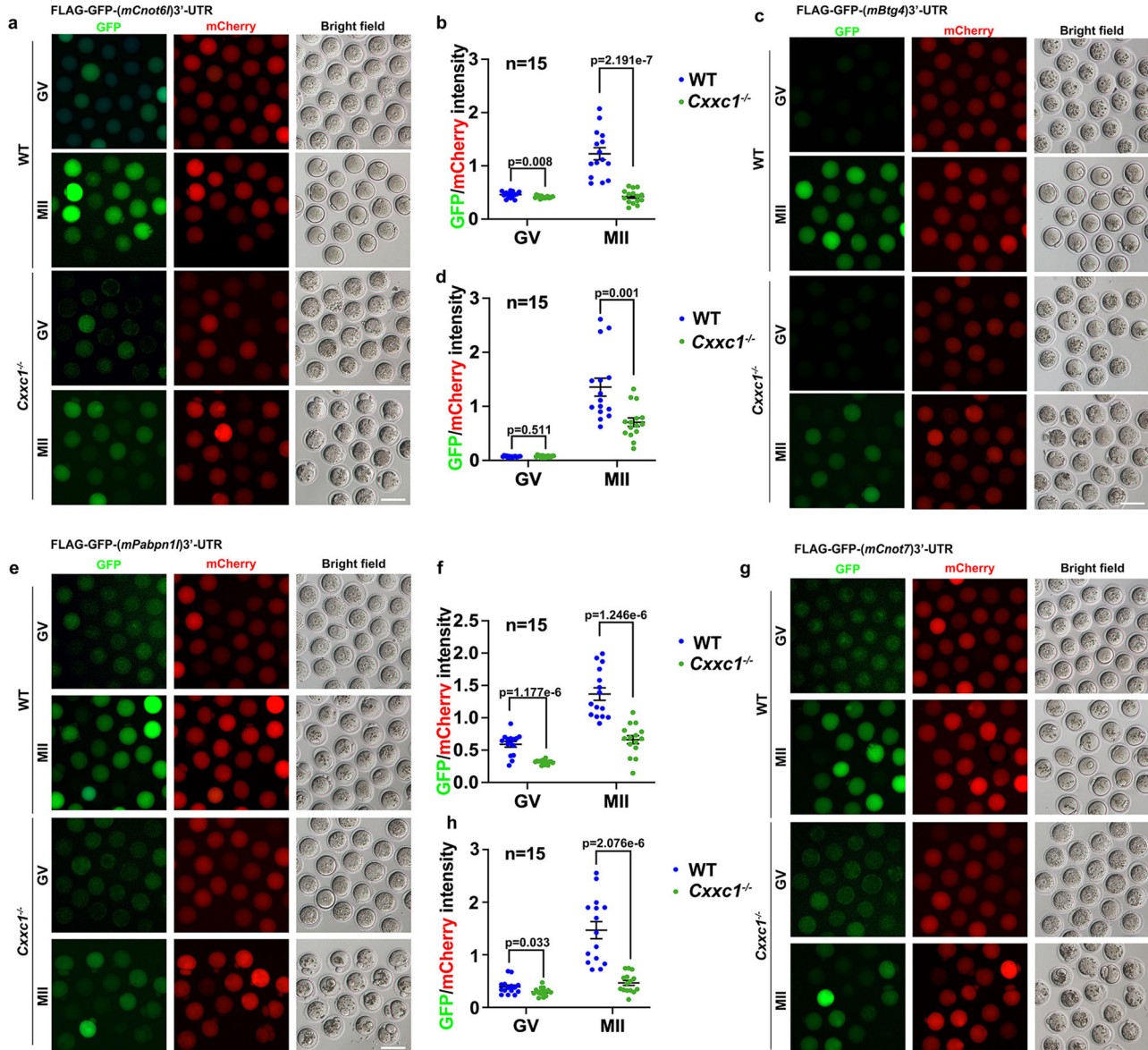

**Fig. 9 Translational activities of transcripts encoding factors related to maternal mRNA clearance in oocytes of 4-week-od WT or *Cxxc1*-deleted mice.**
Fluorescence microscopy results (**a**) and quantification of the fluorescent signals (**b**) showing the translation activities of the *Cnot6l* 3'-UTR in GV or MII oocytes of 4-week-old WT or *Cxxc1*fl/fl;*Gdf9-Cre* mice. **c**, **d** Fluorescence (**c**) and quantification (**d**) showing the translation activities of the *Btg4* 3'-UTR in GV or MII oocytes of 4-week-old WT or *Cxxc1*fl/fl;*Gdf9-Cre* mice. Fluorescence (**e**) and quantification (**f**) showing the translation activities of the *Pabpn1l* 3'-UTR in GV or MII oocytes of 4-week-old WT or *Cxxc1*fl/fl;*Gdf9-Cre* mice. **g**, **h** Fluorescence (**g**) and quantification (**h**) showing the translation activities of the *Cnot7* 3'-UTR in GV or MII oocytes of 4-week-old WT or *Cxxc1*fl/fl;*Gdf9-Cre* mice. Scale bar = 100 μm (**a**, **c**, **e**, **g**). In **b**, **d**, **f** and **h**, data are presented as mean values ± SEM. *P* by two-tailed Student's *t*-test. n.s.: non-significant.

*Cxxc1*-null oocytes (Figs. 7, 8 and 9); 4) Although the overlapped transcripts between aged oocytes and *Cxxc1*-deficient oocytes at the GV stage are limited, the transcripts failed to be degraded and then accumulated in aged oocytes and *Cxxc1*-deficient oocytes at the MII stage showed a remarkable overlap (Fig. 6g). In support of our findings, a recent study analyzed the regulatory layers in multiple mouse and human tissues and discovered that CXXC1 is a conserved modulator at the core of the molecular footprint of aging[45].

Similar to the chromosome cohesion event, epigenetic markers are hieratically established during oogenesis, starting from the embryonic stage in female individuals[46]. H3K4me3 is a crucial junction of the epigenetic network in oocytes[47]. H3K4me3 is deposited on narrow regions of the gene promoter to stimulate

transcription in growing oocytes and then accumulates in broad areas of the gene bodies to facilitate genome transcription silencing in fully grown oocytes[28]. The normal epigenetic landscape maintained by CXXC1 profoundly affects transcriptional activities in the corresponding genomic regions, including genes related to mRNA processing, translational activation, and degradation. Through this mechanism, the epigenetic and cytoplasmic aspects of oocyte maturation are synchronized in the processes of both normal development and aging (Fig. 8).

It would be ideal to investigate the epigenetic landscape of histones in aged oocytes, but this is technically challenging in mammalian species because of the scarcity of mature oocytes in aged individuals. Previous studies have shown that in addition to CXXC1, MLL2 is also required to maintain a normal level of

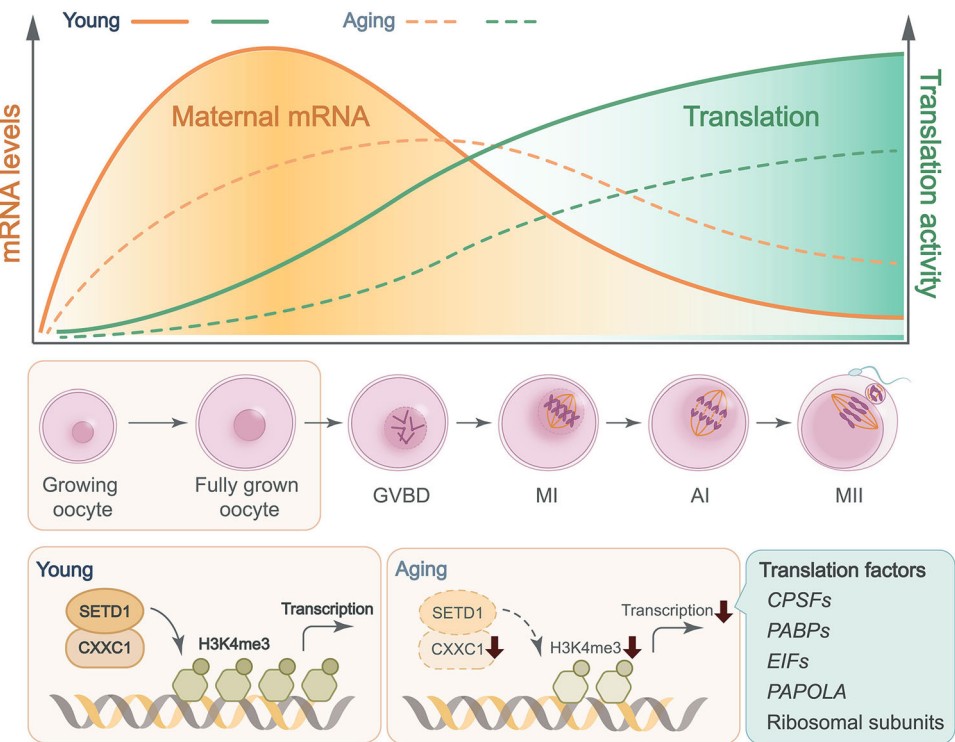

**Fig. 10 CXXC1-maintained histone H3K4 trimethylation and meiosis-coupled mRNA decay mediate age-associated transcriptome changes in human oocytes.** Meiotic maturation-coupled mRNA translation and decay are impaired in the oocytes of aged females. Histone H3K4 trimethylation level was high in fully grown oocytes derived from young women, but gradually decreased during the aging process. The decline of CXXC1-maintained H3K4me3 level led to a wide spectrum of aging-related phenotypes, including aneuploidy after meiotic division, organelle clustering, inefficiency of mRNA transcription, translation, and clearance, anovulation, and ultimately, female infertility.

H3K4 trimethylation in mouse oocytes[48]. CXXC1-SETD1 and MLL2 have partially complementary functions by targeting different areas of the oocyte genome[47,49]. Phenotypes resembling premature oocyte aging were also observed in *Mll2*-deleted oocytes. Therefore, the decreased expression of MLL2 in aged human oocytes may also be partially responsible for age-associated H3K4me3 loss.

A general view is that the mouse is a suitable model system to investigate the molecular basis of oocyte aging in humans. Indeed, many comparable aspects were observed in oocyte aging between mice and humans, although some differences were also observed in our current study. For example, the levels of transcripts encoding mRNA deadenylation factors, including BTG4, CNOT7, and CNOT6L, significantly decreased in oocytes from women over the past 35 years, but not in aged or *Cxxc1^fl/fl;Gdf9-Cre* female mice. A rational explanation is that the reproductive aging process of humans (more than a decade) is much longer than that in mice (a few months), in terms of absolute time. Therefore, the oocyte transcriptome changes in aged mice are relatively less significant than those in humans.

In addition, our study indicated that levels of more transcripts were more downregulated at the GV stage in aged mouse and human oocytes than in a previous study[39–41]. We noticed that the ERCC spike-in was not used in previous studies to normalize the amount of total RNA in the samples. Without normalization using spike-in RNAs, the level changes of many transcripts that fluctuate in proportion with the total RNA amount are likely to be overlooked during analyses. To confirm the importance of proper normalization of transcriptome, we performed differentially expressed genes analysis on our data without ERCC spike-in normalization. The number of differentially expressed genes significantly reduced compared to those with ERCC spike-in

normalization in human oocyte (Fig. S1c, d). Moreover, without ERCC spike-in normalization, the pattern of aging-associated transcriptome changes became unsignificant. Thus, adding spike-in RNAs is important in the study of oocyte transcriptomes, as mature oocytes are transcriptionally silent, and balance between transcription and degradation does not exist. Therefore, our results updated the description of aging-associated transcriptome changes in GV oocytes and emphasized the need for proper normalization in analyzing dynamic transcriptome changes.

Previous research reported that although a number of transcripts were differentially translated, no significant differences in global translational rates were found between young and aged oocytes[11]. Particularly, increased expression of maturation promoting factor components was observed, which leads to accelerated meiosis in oocytes from aged females[50]. In ref. [11] the association of maternal transcripts with polyribosome at the whole transcriptome level was detected. These results are no doubt valuable in evaluating the translation activity of specific transcripts, and also suggested that different transcripts may also have different 3′-UTR-mediated translational efficiencies. In this study, we provided results of reporter assays showing the decreased translation of maternal transcripts encoding factors related to the CCR4-NOT deadenylase complex in aged mouse oocytes. It is possible that this category of maternal transcripts is particularly sensitive to age-associated cytoplasmic changes, because they remain translationally dormant in GV oocytes but need to be promptly recruited by the polyadenylation and translation machinery during meiotic resumption. In consistent with our hypothesis, polysome isolation and sequencing results (Fig. S2d, extracted from original data of ref. [11]) also showed that the subunits of CCR4-NOT is decreased in aged mouse oocytes[11]. Although the standard errors between different samples was too

large to make significant difference, these results suggested decreased translation of these transcripts in aged oocytes.

Clinically, GV stage oocytes collected from patients with declining ovarian status are often cultured for extended periods in medium containing GVBD inhibitory molecules, such as IBMX, an inhibitor of cAMP phosphodiesterase. In this way, nuclear maturation is placed on hold, giving the oocytes more time to achieve optimal cytoplasmic maturation. However, the molecular mechanisms underlying this treatment are not clearly understood. Recent studies, including ours, have suggested that the preparation of cytoplasmic mRNA translation and degradation machinery might be the key event occurring in these meiosis-arrested oocytes. By deliberately delaying meiotic cell cycle progression, maternal transcripts are more thoroughly removed, and the resulting mature oocytes are endowed with better developmental competence.

In summary, this study generated a genome-wide database of meiotic maturation-coupled mRNA degradation in both young and aged mouse and human oocytes. Furthermore, using a gene knockout model, we demonstrated the role of CXXC1-maintained H3K4 trimethylation in reinforcing mRNA translation and degradation activities in maturing mouse oocytes, by which the epigenetic and cytoplasmic aspects of oocyte maturation are coordinated. Due to its physiological importance and the intrinsic vulnerability of maintaining this epigenetic mark in an extended lifespan, the network of CXXC1-maintained H3K4me3 in association with mRNA decay competence sets up a timer for oocyte aging in both mice and humans.

## Methods

**Oocyte collection**. All human oocytes were collected from volunteers. The ovaries were stimulated with GnRH analogues combined with recombinant follicle stimulating hormone (FSH). Oocytes were obtained through follicle puncture 36 h after hCG administration. The cumulus cells around each oocyte were removed using hyaluronidase treatment. 35 female participants were from four different age groups:< 30 years old; 30-35 years old; 36–39 years old; and ≥40 years old. These participants were aware of the research purpose in advance, and voluntarily and free donate oocytes. Informed consents were obtained from all participants. Donated oocytes were randomly selected for investigation after informed consent was obtained in above four age groups. The experiments performed in this study were approved and guided by the ethical committee of the Reproductive Medicine Center of Guangdong Second Provincial General Hospital (Research license 20190906-01-03-YXKYYJ-GZRKT) and the Reproductive & Genetic Hospital of CITIC-XIANGYA (Research license LL-SC-2017-012-1). The study design and conduct complied with all relevant regulations regarding the use of human study participants and was conducted in accordance with the criteria set by the Declaration of Helsinki.

**Animals**. All mouse strains used were constructed in a C57BL6 background. Wild-type C57BL6 female mice were obtained from the Zhejiang Academy of Medical Science, China. $Cxxc1^{fl/fl};Gdf9-Cre$ female mice have been previously reported. Mice were maintained under specific pathogen free (SPF) conditions in a controlled environment at 20–22 °C with a 12/12 h light/dark cycle, 50–70% humidity and food and water provided ad libitum. All experimental protocols involving mice were approved by the Zhejiang University Institutional Animal Care and Research Committee (Approval # ZJU20170014), and mouse care and use were performed in accordance with the relevant guidelines and regulations.

**Oocyte culture**. Mice at 4 weeks of age were injected with 5 IU of PMSG and humanely euthanized 44 h later. Oocytes at the GV stage were harvested in M2 medium (M7167; Sigma-Aldrich) and cultured in mini-drops of M16 medium (M7292; Sigma-Aldrich) covered with mineral oil (M5310; Sigma-Aldrich) at 37 °C in a 5% $CO_2$ atmosphere. In some experiments, milrinone (2 μM) was added to the culture medium to inhibit spontaneous GVBD.

**In vitro transcription and preparation of mRNAs for microinjections**. To prepare mRNAs for microinjection, expression vectors were linearized and subjected to phenol/chloroform extraction and ethanol precipitation. The linearized DNAs were transcribed in vitro using the SP6 message mMACHINE Kit (Invitrogen, AM1340). Poly(A) tails (~200–250 bp) were added to transcribed mRNAs using the Poly(A) tailing kit (Invitrogen, AM1350) and were recovered by lithium chloride precipitation and resuspended in nuclease-free water.

**Microinjection of oocytes**. For microinjection, fully grown GV oocytes were harvested in M2 medium with 2 μM milrinone to inhibit spontaneous GVBD. All injections were performed using an Eppendorf Transferman NK2 micromanipulator. Denuded oocytes were injected with 5–10pL samples per oocyte. The concentration of all injected RNAs was adjusted to 500 ng/μL. After injection, oocytes were washed and cultured in M16 medium at 37 °C with 5% $CO_2$.

**Immunofluorescence and confocal microscopy**. Oocytes were fixed with 4% paraformaldehyde in PBS. The cells were permeabilized with 0.3% Triton X-100 in PBS. Antibody staining was performed using standard protocols, as previously described[51]. The antibodies used in the experiments are described in Supplementary Data 3. Imaging was performed using a Zeiss LSM710 confocal microscope. Quantitative analysis of the fluorescence signals was conducted using the NIH ImageJ software, as previously described[52].

**ROS detection**. ROS were detected using the ROS detection assay kit (Beyotime) according to the manufacturer's instructions. In brief, oocytes were stained with 2',7'-dichlorofluorescin diacetate (DCFH-DA) in M2 medium for 20 min at 37 °C, washed, mounted on a glass slide, and examined under a confocal laser scanning microscope (Zeiss LSM 710, Carl Zeiss AG, Germany).

**Mitochondrial membrane potential assay**. The mitochondrial membrane potential assay was performed using the assay kit with JC-1 (Beyotime) according to the manufacturer's instructions. Briefly, oocytes were stained with JC-1 in M2 medium for 20 min at 37 °C, washed with JC-1 buffer, mounted on a glass slide, and examined under a confocal laser scanning microscope (Zeiss LSM 710, Carl Zeiss AG, Germany).

**Western blot analysis**. Oocytes were lysed in protein loading buffer and heated at 95 °C for 5 min. SDS-PAGE and immunoblotting were performed following standard procedures using a Mini-PROTEAN Tetra Cell System (Bio-Rad, Hercules, CA, USA). The primary antibodies and dilution factors used are listed in Supplementary Data 3.

**Detection of protein synthesis**. Oocytes were incubated in M16 medium containing 100 mM L-homopropargylglycine (HPG; a methionine analog that is incorporated into nascent proteins during active protein synthesis) for 1 h and then fixed for 30 min at room temperature in 4% paraformaldehyde (PFA). HPG signals were detected using a Click-iT®HPG Alexa Fluor®Protein Synthesis Assay Kit (Life Technologies). Mean intensity of the HPG signal was measured across the middle of each oocyte and quantified using ImageJ software[53].

**RNA isolation and real-time RT-PCR**. Oocytes or embryos were collected and lysed in 2 μl of lysis buffer (0.2% Triton X-100 and 4 IU RNase inhibitor), followed by reverse transcription with primer transcript II reverse transcriptase (Takara), according to the manufacturer's instructions. Real-time RT-PCR analysis was performed using the LightCycler 480 SYBR Green I Master (Roche) and a Roche 480 Real-Time PCR System. The respective cycle threshold (Ct) values were obtained, and relative mRNA levels were calculated by normalization to endogenous $Gapdh$ mRNA levels (internal control) using Microsoft EXCEL®. Gene expression levels were calculated using $2^{\Delta Ct}$ ($2^{\Delta Ct \ (genes- \ Gapdh)}$). The relative transcript levels of the samples were compared to those of the controls, and fold changes were determined. For each experiment, qPCR was performed in triplicate. The primer sequences used are listed in Supplementary Data 4.

**RNA-seq library preparation**. Oocytes and zygotes were collected from the indicated genotypes (10 oocytes or embryos per sample). Each sample was lysed directly with 4 μl lysis buffer (0.2% Triton X-100, RNase inhibitor, dNTPs, oligo-dT primers, and 0.5 μl 1:10000 ERCC mRNA spike-in dilution for human oocyte samples and 0.2 μl 1:1000 ERCC mRNA spike-in dilution for mouse oocyte samples) and immediately used for cDNA synthesis using the Smart-seq2 method as previously described[54].

Sequencing libraries were constructed from 500 pg of amplified cDNA using TruePrep DNA Library Prep Kit V2 for Illumina (Vazyme, TD503) according to the manufacturer's instructions. Barcoded libraries were pooled and sequenced on the Illumina HiSeq X Ten platform with 150 bp paired-end reads.

**RNA-Seq data analysis**. RNA-seq was performed using biological replicates for all samples. Raw reads were trimmed with Trimmomatic (v0.36) to 50 bp and mapped to the mouse genome (mm10) using TopHat (v2.0.11) with default parameters. Uniquely mapped reads were subsequently assembled into transcripts guided by reference annotation (University of California at Santa Cruz [UCSC] gene models) with Cufflinks (v2.2.1). The expression level of each gene was quantified as the fragments per kilobase of transcript per million mapped reads (FPKM) and was further normalized with the ERCC spike-in. Samples prepared in different batches were normalized to the WT GV-stage oocyte samples in each batch. Only expressed genes (FPKM > 1 in at least one sample) were considered in all analyses,

unless otherwise specified. Functional annotation was performed using DAVID (https://david.ncifcrf.gov/).

Statistical analyses were performed using R software (http://www.rproject.org). The Spearman correlation coefficient ($r_s$) was calculated using the cor function, and the complete linkage hierarchical algorithm was used to cluster the genes. Quality controls of RNA-seq results are provided in Supplementary Data 5-6.

**Statistics and reproducibility**. Results are presented as means ± S.E.M. Most experiments included at least three independent samples. Western blot were repeated at least three times with similar results. Results for two experimental groups were compared using two-tailed unpaired Student's t-tests. The statistically significant values of P is calculated by two-tailed Student's t-test.

**Reporting summary**. Further information on research design is available in the Nature Research Reporting Summary linked to this article.

## Data availability

The data supporting the findings of this study are available from the corresponding authors upon reasonable request. RNA-seq data have been deposited in the NCBI Gene Expression Omnibus database under the accession code GSE173598. Source data for the figures and supplementary figures are provided as a Source Data file.

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

## Acknowledgements

This study was supported by the National Key Research and Developmental Program of China [2021YFC2700100 to Q.Q.S. and H.Y.F., 2021YFA1100300, SQ2020YFF0426502 to Q.Q.S.], National Natural Science Foundation of China [91949108 to Q.Q.S., 31930031 to H.Y.F.], Outstanding Youth Foundation of Guangdong Province [2022B1515020038 to Q.-Q.S.], the Key Research and Development Program of Zhejiang Province [2021C03098, 2021C03100 to H.Y.F.]. and Start-up Funding of Guangdong Second Provincial General Hospital [YY2019-001 to Q.-Q.S.].

## Author contributions

Q.Q.S. and H.Y.F. conceived, designed, and supervised the work. Q.Q.S. and H.Y.F. wrote the manuscript. Y.W.W. performed human and mouse oocyte RNA-seq experiments, respectively. Y.W.W. and Q.Q.S analyzed the RNA-seq data and generated the figures. S. L. and W. Z. collected equal amounts of human oocytes with the help of G.L. used in this study. L.C. collected mouse oocytes used for RNA-seq. Y.C.L. and Z.Q.D. performed mouse genotyping. Q.Q.S. performed, Y.W.W. and Y.Z. helped with the immuno-fluorescence, and Y.C.L. helped with real-time PCR of mouse oocytes.

## Competing interests

The authors declare no competing interests.
