## [Peer Review File · Nature Communications]

Title: Dynamic mRNA Degradome Analyses Indicate a Role of Histone H3K4 Trimethylation in Association with Meiosis-coupled mRNA Decay in Oocyte AgingReviewers' comments:

Reviewer #1 (Remarks to the Author):

The manuscript by Wu et al explores the relationship between chromatin modification (H3K4me3) and meiotic maturation-coupled mRNA degradation during ageing. The authors quantify the age-related changes to the human GV and MII transcriptomes and make some important novel observations. The authors demonstrate that human aged oocytes overall have reduced amount of mRNA with a significant decline in mRNAs that encode for RNA degradation and the COMPASS H3K4me3 HMTase complex. They demonstrate that meiotic maturation-coupled mRNA degradation is impaired in aged human oocytes. A similar phenomenon was observed in mice. Furthermore, they find a reduction of CXXC1 and H3K4me3 during oocyte ageing. Deletion of CXXC1 resulted in a premature oocyte ageing phenotype that share some of the transcriptomic changes associated with ageing. Using a single reporter assay of the RNA degradation factor Btg4 the authors demonstrate that aged oocytes have reduced translation of GFP when coupled to the 3'UTR of this factor. In summary the authors identify defective meiotic maturation-coupled mRNA degradation as a key cytoplasmic event as well as reduced H3K4me3 that contribute to oocyte ageing. Overall, most of the key conclusions are well supported by the data in the manuscript and I am supportive of publication. I have one major concern (point 2 below) that would need to be addressed first.

Major points

1. Figure 1A and 2A would need some statistical evaluation to understand the significance between the various age groups.
2. In my opinion Figure 7 is the weak point of the manuscript. I am not sure that one can make such a broad conclusion from a single reporter assay. I suggest that if the authors wish to incorporate defective translation into their model in Figure 8 then this aspect would need to be better studied. This could be done by performing Ribo-seq from young and aged oocytes; as well as using other methods such as methionine or OP-puro labelling to measure translational output.

Reviewer #2 (Remarks to the Author):

In the manuscript titled "Dynamic mRNA Degradome Analyses Indicate a Role of Histone H3K4 Trimethylation in Association with Meiosis-coupled mRNA Decay in Oocyte Aging", Wu et al. compared the transcriptome between young and aged GV and MII oocytes from human and mice. They found that maternal mRNA storage decreased in aged GV oocytes but increased in aged MII oocytes compared with young oocytes in humans. Further analyses indicated that the degradation of maternal mRNAs is impaired in the aged oocytes. Next, their results point to a potential association between CXXC1 mediated H3K4me3 and oocytes aging. Cxxc1 knockout in mouse oocytes leads to defects in mRNA

decay similar to that observed in aged oocytes. Lastly, they provide evidence that CXXC1 mediated H3K4me3 contributes to the translation of some maternal factors and is required for the M-decay of maternal mRNAs. The authors propose an interesting link between histone modification, maternal mRNA degradation, and oocyte aging. However, I found that some conclusions have been mentioned in previous studies, which limits its novelty: transcriptome analyses of young and aged GV and MII oocytes in humans and mice had been reported previously in several literatures, and the disturbing mRNA degradation in aged MII oocytes has also been shown in mice (PMID: 18342300). And more importantly, the authors need to provide additional data to demonstrate that CXXC1-maintained H3K4me3 plays a leading role in oocyte aging in both mice and humans.

Major concerns:

Line 65, the authors described “A recent study demonstrated that decreased mRNA translation activity is a hallmark of oocyte aging in terms of cytoplasmic maturation (11).”. However, this cited paper and previous research (PMID: 30235877) reported that although a number of transcripts were differentially translated, no significant differences in global translational rates were found between young and aged oocytes. Thus, the general protein translation activity decreased with aging in maturing mouse oocytes should not be considered as a hallmark for oocyte aging. The author should discuss these references when report their observation of decreased general protein translation in aging oocytes.

Line 285, “Compared to WT, 240 and 1013 transcripts were increased or decreased more than 5-fold in Cxhc1 null oocytes at the GV stage, respectively (Fig. 6A).”. In Fig. 6A, it seems more up-regulated transcripts than down-regulated transcripts present in Cxhc1 null oocytes at the GV stage. Please confirm the number of transcripts in Fig. 6A is correct.

Line 288, the author described “The total mRNA levels decreased in Cxhc1 null oocytes at the GV stage, but were higher than those in WT oocytes at the MII stage, apparently due to insufficient degradation (Fig. 6B).”. In Fig.6B, the total mRNA levels appear to be increased in Cxhc1 null oocytes at the GV stage. Please clarify this.

In Fig. 6c, more than a thousand transcripts were dysregulated in aged oocytes or Cxhc1-deficient oocytes, but the overlap between them is limited. The authors need to provide more data to demonstrate that the Cxhc1-deficient oocytes resemble aged oocytes.

Line 302, the section titled “CXXC1 maintains cytoplasmic translation of maternal factors required for mRNA decay”, but I couldn’t find the data directly support this claim. The authors only demonstrated that the general protein translation activity is decreased with aging in Fig.4, but no result in Cxhc1 null oocytes is shown.

In Fig. 6, translation of Btg4 in oocytes is highly dependent on CPSF, which expression level should be analyzed. Additional reporter assay should be performed to detect the translation of other maternal factors required for mRNA decay, such as CNOT7 and CNOT6L.

The authors found that in aged human GV oocytes, the mRNA showed decreased expression levels compared to young oocytes (Fig 1A and Fig.2E), but they also showed that the mRNA level are comparable in young and aged mouse GV oocytes (Fig. 3C and F). The model presents in Fig.8 should be modified.

The authors should check the citation carefully throughout their manuscript. Line 73, the author should cite the more original researches on the CCR4-NOT complex and another simultaneously published research about BTG4 (PMID: 27190313). Line 389, "In addition, our study indicated that levels of more transcripts were more downregulated at the GV stage in aged mouse and human oocytes than in a previous study (35-37)". Human MII but not GV oocytes were used in ref. 35.

Minor concerns:

The statistical significance should be added in Fig.1A, Fig.2B, and Fig.2C.

Explain the reason for using "FPKM + 1" in Fig.1 and Fig.2, but "FPKM + 0.1" in Fig.3 and Fig.6.

Line 160-162, the authors described that "The decrease in transcripts encoding regulators of mRNA stability drew our particular notice as recent studies have indicated that 1) mutations in these factors cause the zygotic arrest and female infertility in humans and mice....." and cited the paper of BTG4. However, BTG4 is not listed in the mRNA decay categories in Fig.1E.

The authors should define "M-decay".

Line 203, the authors described "we isolated RNA from these oocytes and performed RNA sequencing". In the materials and methods sections, the authors described the RNA-seq libraries were prepared directly using the oocyte lysate.

The supplementary table S1-6 should be supplied.

Line 264, "Cxxc1 mRNA and protein levels were downregulated in aged mouse and human oocytes". The authors only displayed the decreased mRNA levels in human oocytes and the decreased protein levels in mouse oocytes. The data of Cxxc1 mRNA level in mouse oocytes should be shown.

Line 303, the author described "the mRNA levels of BTG4, CNOT7, and CNOT6L did not decrease with advanced age, as detected by RT-qPCR". Are these results also detectable and confirmed by RNA-seq data in aged mouse oocytes?

Line 384, the author claimed that the levels of BTG4, CNOT7, and CNOT6L significantly decreased in oocytes from women over the past 35 years. I could not find the relevant data in the manuscript.

The ">=" in figures should be "≥". This should be corrected throughout the figures.

Line 703, “and Cxhc1-deleted oocytes before and after in vitro maturation culture” should be removed.

Explain “HPG incorporation assay” in the materials and methods section.

Responses to Reviewers' comments (NCOMMS-21-30120):

Reviewer #1 (Remarks to the Author):

The manuscript by Wu et al explores the relationship between chromatin modification (H3K4me3) and meiotic maturation-coupled mRNA degradation during ageing. The authors quantify the age-related changes to the human GV and MII transcriptomes and make some important novel observations. The authors demonstrate that human aged oocytes overall have reduced amount of mRNA with a significant decline in mRNAs that encode for RNA degradation and the COMPASS H3K4me3 HMTase complex. They demonstrate that meiotic maturation-coupled mRNA degradation is impaired in aged human oocytes. A similar phenomenon was observed in mice. Furthermore, they find a reduction of CXXC1 and H3K4me3 during oocyte ageing. Deletion of CXXC1 resulted in a premature oocyte ageing phenotype that share some of the transcriptomic changes associated with ageing. Using a single reporter assay of the RNA degradation factor Btg4 the authors demonstrate that aged oocytes have reduced translation of GFP when coupled to the 3'UTR of this factor.

In summary the authors identify defective meiotic maturation-coupled mRNA degradation as a key cytoplasmic event as well as reduced H3K4me3 that contribute to oocyte ageing. Overall, most of the key conclusions are well supported by the data in the manuscript and I am supportive of publication. I have one major concern (point 2 below) that would need to be addressed first.

Response: We thank the reviewer for carefully review our manuscript and his/her supportive comment that the key conclusions are well supported by the data in the manuscript. We have carefully addressed the reviewer's concern as detailed below.

Major points

1. Figure 1A and 2A would need some statistical evaluation to understand the significance between the various age groups.

Response: We have made statistical evaluations for Figure 1A and 2A as suggested.

2. In my opinion Figure 7 is the weak point of the manuscript. I am not sure that one can make such a broad conclusion from a single reporter assay. I suggest that if the authors wish to incorporate defective translation into their model in Figure 8 then this aspect would need to be better studied. This could be done by performing Ribo-seq from young and aged oocytes; as well as using other methods such as methionine or OP-puro labelling to measure translational output.

Response: We appreciate the reviewer's comments. However, due to technical obstacles Ribo-seq is not applicable to a small number of cells at current stage. So we can't perform Ribo-seq from young and aged oocytes;

Dr. Andrej Susor's group had already reported that genome-wide translome profiling in young and old mouse oocytes based on polysome isolation and sequencing (Del Llano, E., et al., Aging Cell 2020). This study reveals considerable numbers of transcripts that are differentially translated in oocytes obtained from aged compared to young females. A number of aberrantly translated mRNAs in oocytes from aged

females are associated with spindle assembly and cell division.

In addition, in this study, we focus on the important maternal factors. as also suggested by reviewer 2, we performed more reporter assays to detect the translation of other important maternal factors required for meiotic maturation and mRNA degradation, including PABPN1L, CNOT7, and CNOT6L. The new results in Fig. 8 and 9 showed that the meiotic maturation-associated translational activation of these transcripts was compromised in oocytes from WT mice with advanced age and *Cxxc1* conditional KO mice.

Collectively, these results all suggested defective translation of maternal mRNAs required for mRNA degradation in aged oocytes.

Reviewer #2 (Remarks to the Author):

In the manuscript titled “Dynamic mRNA Degradome Analyses Indicate a Role of Histone H3K4 Trimethylation in Association with Meiosis-coupled mRNA Decay in Oocyte Aging”, Wu et al. compared the transcriptome between young and aged GV and MII oocytes from human and mice. They found that maternal mRNA storage decreased in aged GV oocytes but increased in aged MII oocytes compared with young oocytes in humans. Further analyses indicated that the degradation of maternal mRNAs is impaired in the aged oocytes. Next, their results point to a potential association between CXXC1 mediated H3K4me3 and oocytes aging. *Cxxc1* knockout in mouse oocytes leads to defects in mRNA decay similar to that observed in aged oocytes. Lastly, they provide evidence that CXXC1 mediated H3K4me3 contributes to the translation of some maternal factors and is required for the M-decay of maternal mRNAs. The authors propose an interesting link between histone modification, maternal mRNA degradation, and oocyte aging.

Response: We highly appreciate the reviewer’s insightful, specific, and constructive comments and suggestions to our manuscript. We thank the reviewer for pointing out that the manuscript proposes an interesting link between histone modification, maternal mRNA degradation, and oocyte aging. We have made extensive efforts to provide more experimental details, elaborate on our data analyses, interpretations, and conclusions, and improve the quality of the content during revision.

However, I found that some conclusions have been mentioned in previous studies, which limits its novelty: transcriptome analyses of young and aged GV and MII oocytes in humans and mice had been reported previously in several literatures, and the disturbing mRNA degradation in aged MII oocytes has also been shown in mice (PMID: 18342300). And more importantly, the authors need to provide additional data to demonstrate that CXXC1-maintained H3K4me3 plays a leading role in oocyte aging in both mice and humans.

Response: We appreciate the reviewer’s comments that point out the weakness and limitations of our manuscript. We carefully checked these references and would like to politely provide some explanations:

1) Previous studies analyzed mouse and human oocyte transcriptome in correlation

with advanced maternal age mainly focused on the two meiotic arrest points, GV and/or MII stages [ref. 1-3]. These transcriptome analyses, although undoubtful valuable at the time, might not reflect the dynamic transcriptome changes, or mRNA degradome, when comparing the young and aged oocytes. To our knowledge, we are the first to purposely interrogate the whole mRNA degradome in human oocytes of different maternal ages.

2) In the paper mentioned by reviewer 2 (PMID: 18342300; [ref. 3]), the authors only observed that “approximately 5% of the transcripts are differentially expressed in oocytes obtained from old females when compared to oocytes obtained from young females”. They detected the transcriptome changes in mouse oocytes using microarray, which was a state-of-the-art technique at that time but is less sensitive than RNA-seq. Nor did they propose any potential mechanisms underlying these age-associated changes. In contrast, our study proposed a potential link between histone modification and oocyte aging. We demonstrated the role of CXXC1-maintained H3K4 trimethylation in reinforcing mRNA translation and degradation activities in maturing mouse oocytes, by which the epigenetic and cytoplasmic aspects of oocyte maturation are coordinated.

3) These studies only described transcriptome changes in mouse OR human oocytes. Our study compared the age-associated transcriptome changes in mouse AND human, and observed more serious mRNA decay defects in the aged human oocytes than in the aged mouse oocytes.

4) The transcriptome analyses in these previous studies may have a few limitations: (i) The ERCC spike-in was not used to normalize the amount of total RNAs in the samples. Without normalization using spike-in RNAs, the level changes of many transcripts that fluctuate in proportion with the total RNA amount are likely to be overlooked during analyses; (ii) In these studies, only two groups (young and aged) were included in the analyses. In contrast, we analysed the gradual transcriptome changes during the whole human reproductive life, and described the key transition point at which oocyte quality declines.

We thank the reviewer again for these helpful critics and have included these explanations in the Discussion.

Major concerns:

Line 65, the authors described “A recent study demonstrated that decreased mRNA translation activity is a hallmark of oocyte aging in terms of cytoplasmic maturation (11).”. However, this cited paper and previous research (PMID: 30235877) reported that although a number of transcripts were differentially translated, no significant differences in global translational rates were found between young and aged oocytes. Thus, the general protein translation activity decreased with aging in maturing mouse oocytes should not be considered as a hallmark for oocyte aging. The author should discuss these references when report their observation of decreased general protein translation in aging oocytes.

Response: We sincerely thank the reviewer for pointing out our incorrect description of the published studies. We have revised the description of these cited papers, and discussed the potential reasons for the discrepancy between these research and our

results. In Ref 11, the authors detected the association of maternal transcripts with polyribosome at the whole transcriptome level. These results are no doubt valuable in evaluating the translation activity of specific transcripts, but may not directly reflect the global protein synthesis activity because different transcripts have different abundance in oocytes and may also have different 3'-UTR-mediated translational efficiencies. Therefore, in our opinion, the polysome sequencing results may not completely similar with the measurement of global protein synthesis activity. Furthermore, we provided more experimental results of reporter assays showing the decreased translation of maternal transcripts encoding factors related to the CCR4-NOT deadenylase complex. It is possible that this category of maternal transcripts is particularly sensitive to age-associated cytoplasmic changes, because they remain translationally dormant in GV oocytes but need to be promptly recruited by the polyadenylation and translation machinery during meiotic resumption. As the reviewer suggested, we discussed these references when reporting our observation of decreased protein translation in aging oocytes.

Line 285, "Compared to WT, 240 and 1013 transcripts were increased or decreased more than 5-fold in *Cxhc1* null oocytes at the GV stage, respectively (Fig. 6A)". In Fig. 6A, it seems more up-regulated transcripts than down-regulated transcripts present in *Cxhc1* null oocytes at the GV stage. Please confirm the number of transcripts in Fig. 6A is correct.

Response: We thank the reviewer for pointing out this issue. The reason that caused "it seems more up-regulated transcripts than down-regulated transcripts present in *Cxhc1* null oocytes at the GV stage" is the existence of those low-transcript-level genes. Among those down-regulated genes in *Cxhc1* null oocytes at the GV stage, transcription level of 774 genes were down-regulated to 0 by FPKM measurement, thus dots representing these genes were concentrated to line: $y = \log_2(0+0.1) = -0.3$, which was hard to be noticed in the original Fig. 6A. Now we have adjusted the FPKM+0.1 to FPKM+1 in the new submission to avoid the misunderstanding.

Line 288, the author described "The total mRNA levels decreased in *Cxhc1* null oocytes at the GV stage, but were higher than those in WT oocytes at the MII stage, apparently due to insufficient degradation (Fig. 6B)". In Fig.6B, the total mRNA levels appear to be increased in *Cxhc1* null oocytes at the GV stage. Please clarify this.

Response: We thank the reviewer for pointing out this issue. The original version contained a total of 55477 annotated genes, among which, 28915 genes barely have no expression level in all samples. The presence of nearly 50% extremely low-expression-level genes lowered the overall expression level and made the difference less significant between groups. Considering those extremely low-expression-level genes are biologically meaningless, we removed them and redrew the box diagram in Fig.6B. The new diagram in Fig.6B shows an obvious trend that total mRNA levels decreased in *Cxhc1* null oocytes, compared with WT oocytes at the GV stage.

In Fig. 6c, more than a thousand transcripts were dysregulated in aged oocytes or

Cxxc1-deficient oocytes, but the overlap between them is limited. The authors need to provide more data to demonstrate that the Cxxc1-deficient oocytes resemble aged oocytes.

Response: We agree with the reviewer that the overlapped transcripts between aged oocytes and Cxxc1-deficient oocytes are limited. As the reviewer suggested, we emphasized other evidence to demonstrate that the Cxxc1-deficient oocytes resemble aged oocytes. These include:

1) Histone H3K4me3 level significantly decreased in both aged human and mouse oocytes. This epigenetic change resembles what happened in Cxxc1-deficient oocytes. Meanwhile, Cxxc1 expression level also decreased in aged mouse oocytes. This observation suggested the possibility that Cxxc1-deficient mouse oocytes may display some similar phenotypes as the aged oocytes;

2) Some cytoplasmic changes, including increases of ROS levels, aggregation of mitochondria, decreases of mitochondrial DNA copy numbers as well as mitochondrial membrane potential, were observed in both aged oocytes and Cxxc1-null oocytes, suggesting that Cxxc1 deletion caused premature aging in mouse oocytes (Fig. 5).

3) At the molecular level, we demonstrated that meiotic maturation-associated mRNA decay was impaired in both aged oocytes (Fig. 3B, C, F) and Cxxc1-null oocytes (Fig. 6A, B, E, F). This is due to the inefficient translation of transcripts encoding mRNA turnover factors (*Btg4*, *Pabpn1l*, *Cnot6l*) in both aged oocytes and Cxxc1-null oocytes (revised Fig. 7 and 8).

4) We also adjusted Fig. 6C. When we selected the transcripts of FPKM>1 in WT oocytes for analyses, the overlapped transcripts between aged oocytes and Cxxc1-deficient oocytes increased. Therefore, we speculated that the overlap between them might be limited by some transcripts expressed at very low levels. Nevertheless, the level changes of these sparsely expressed transcripts may have little effect on the function of oocytes.

5) Although the overlapped transcripts between aged oocytes and Cxxc1-deficient oocytes at the GV stage are limited, the transcripts failed to be degraded and then accumulated in aged oocytes and Cxxc1-deficient oocytes at the MII stage showed a remarkable overlap (Fig. 6G).

We have added these explanations in the revised Discussion.

Line 302, the section titled “CXXC1 maintains cytoplasmic translation of maternal factors required for mRNA decay”, but I couldn't find the data directly support this claim. The authors only demonstrated that the general protein translation activity is decreased with aging in Fig.4, but no result in Cxxc1 null oocytes is shown.

Response:

We thank the reviewer for pointing out the weakness of the original manuscript. We performed new experiments to analyze the maternal mRNA translational activity in Cxxc1 null oocytes, and provided new results to support the conclusion that “CXXC1 maintains cytoplasmic translation of maternal factors required for mRNA decay”.

1) Maternal mRNA translation is highly dependent on CPSF. *Cpsf4* and *Cpsf4l* are highly expressed in oocytes. We analyzed the mRNA expression levels of *Cpsf4* and

Cpsf4l using RT-qPCR, and protein level of CPSF4 by immunofluorescence, respectively. The results showed that the mRNA level of *Cpsf4* have no significant difference between WT and *Cxxc1* null oocytes, and *Cpsf4l* mRNA level is significantly decreased in *Cxxc1* null oocytes. Meanwhile, the immunofluorescence showed that the protein level of CPSF4 is significantly decreased in *Cxxc1* null oocytes at GV and MII stage. The results are presented in the revised Supplementary Figure S3.

2) As the reviewer suggested, reporter assays were performed in *Cxxc1* null oocytes to detect the translation of other maternal factors required for mRNA decay, including BTG4, PABPN1L, MOS, and CNOT6L. The new results in Fig. 8 and 9 showed that the meiotic maturation-associated translational activation of these transcripts was compromised in oocytes from WT mice with advanced age and *Cxxc1* conditional KO mice. Collectively, these results all suggested defective translation of maternal mRNAs required for mRNA degradation in aged oocytes.

In Fig. 6, translation of *Btg4* in oocytes is highly dependent on CPSF, which expression level should be analyzed. Additional reporter assay should be performed to detect the translation of other maternal factors required for mRNA decay, such as CNOT7 and CNOT6L.

Response:

1) We analyzed the mRNA expression levels of multiple CPSF subunits. *Cpsf4* and *Cpsf4l* are highly expressed in oocytes. We analyzed the mRNA expression levels of *Cpsf4* and *Cpsf4l* using RT-qPCR, and protein level of CPSF4 through immunofluorescence, respectively. The results showed that the mRNA level of *Cpsf4* have no significant difference between young and aged oocytes, and *Cpsf4l* mRNA level is significantly decreased in aged oocytes. However, the immunofluorescence showed that the protein level of CPSF4 is significantly decreased in aged oocytes at GV and MII stage (revised Supplementary Figure S3).

2) Additional reporter assays were performed to detect the translation of other maternal factors required for mRNA decay, including PABPN1L, MOS, CNOT7, and CNOT6L. The results in revised Figure 8 showed that translation of all these mRNAs were decreased at MII stage with advanced age.

3) Polysome isolation and sequencing (Del Llano, E., et al., Aging Cell 2020) results also showed that the translation of mRNAs encoding BTG4 and CCR4-NOT subunits decreased in aged mouse oocytes. Although the standard errors between different samples was too large to make significant difference, these results suggested decreased translation of these transcripts in aged oocytes. We have included these analyses in the revised Supplementary Figure S2D.

The authors found that in aged human GV oocytes, the mRNA showed decreased expression levels compared to young oocytes (Fig 1A and Fig.2E), but they also showed that the mRNA level are comparable in young and aged mouse GV oocytes (Fig. 3C and F). The model presents in Fig. 8 should be modified.

Response: We have modified the model presents in Fig. 10. The revised Fig. 10 only

described the mRNA level, translation, and degradation changes in young and old human oocytes during meiotic maturation.

The authors should check the citation carefully throughout their manuscript. Line 73, the author should cite the more original researches on the CCR4-NOT complex and another simultaneously published research about BTG4 (PMID: 27190313). Line 389, “In addition, our study indicated that levels of more transcripts were more downregulated at the GV stage in aged mouse and human oocytes than in a previous study (35-37)”. Human MII but not GV oocytes were used in ref. 35.

Response: We have carefully checked the citation throughout the manuscript as the reviewer suggested. We have cited more original research reporting the role of CCR4-NOT complex in mouse oocytes and other published research about BTG4. We realized that human MII but not GV oocytes were used in ref. 35, and have removed this reference from this location.

Minor concerns:

The statistical significance should be added in Fig.1A, Fig.2B, and Fig.2C.

Response: The statistical significance was added in Fig.1A, Fig.2B, and Fig.2C as suggested.

Explain the reason for using “FPKM + 1” in Fig.1 and Fig.2, but “FPKM + 0.1” in Fig.3 and Fig.6.

Response: In response to the reviewer’s comments, we used “FPKM + 1” in all analyses of RNA-seq results. This application does not affect the conclusions in Fig. 3 and Fig. 6.

Line 160-162, the authors described that “The decrease in transcripts encoding regulators of mRNA stability drew our particular notice as recent studies have indicated that 1) mutations in these factors cause the zygotic arrest and female infertility in humans and mice.....” and cited the paper of BTG4. However, BTG4 is not listed in the mRNA decay categories in Fig.1E.

Response: We are sorry for the oversight. As the reviewer suggested, we have listed *BTG4* in the mRNA decay category in revised Fig. 1E. The results showed that *BTG4* mRNA levels decreased in oocytes from women more than 35 years old.

The authors should define “M-decay”.

Response: We have defined “M-decay” in the revised Introduction.

Line 203, the authors described “we isolated RNA from these oocytes and performed RNA sequencing”. In the materials and methods sections, the authors described the RNA-seq libraries were prepared directly using the oocyte lysate.

Response: In the Smart-seq 2 method, RNA-seq libraries were indeed prepared directly using the oocyte lysate. We are sorry for the wrong description in the original text and have made corrections during revision.

The supplementary table S1-6 should be supplied.

Response: We are sorry for the oversight and supplied these tables in the re-submission.

Line 264, “Cxxc1 mRNA and protein levels were downregulated in aged mouse and human oocytes”. The authors only displayed the decreased mRNA levels in human oocytes and the decreased protein levels in mouse oocytes. The data of Cxxc1 mRNA level in mouse oocytes should be shown.

Response: The data of Cxxc1 mRNA level determined by quantitative RT-PCR in young and old mouse oocytes were shown in revised Fig. 4F. The result showed that Cxxc1 mRNA level between young and old mouse oocytes have no significant difference.

Line 303, the author described “the mRNA levels of BTG4, CNOT7, and CNOT6L did not decrease with advanced age, as detected by RT-qPCR”. Are these results also detectable and confirmed by RNA-seq data in aged mouse oocytes?

Response: We have extracted the mRNA levels of *Btg4*, *Cnot7*, and *Cnot6l* from RNA-seq data of young and aged mouse oocytes (Fig. S2C). Consistent with the RT-qPCR results, the RNA-seq results also indicated that the mRNA levels of BTG4, CNOT7, and CNOT6L did not decrease with advanced age.

Line 384, the author claimed that the levels of BTG4, CNOT7, and CNOT6L significantly decreased in oocytes from women over the past 35 years. I could not found the relevant data in the manuscript.

Response: We are sorry for the oversight. We have included the expression levels of BTG4, CNOT7, and CNOT6L in the revised Fig. 1E. The RNA-seq results showed that the levels of *BTG4*, *CNOT7*, and *CNOT6L* decreased in oocytes from women over 35 years old.

The “>=” in figures should be “≥”. This should be corrected throughout the figures.

Response: We have made the correction as suggested.

Line 703, “and Cxxc1-deleted oocytes before and after in vitro maturation culture” should be removed.

Response: We have made the correction as suggested.

Explain “HPG incorporation assay” in the materials and methods section.

Response: We have supplied the description for “HPG incorporation assay” in the revised “Materials and Methods” section.

REVIEWERS' COMMENTS

Reviewer #1 (Remarks to the Author):

My concerns have been addressed. I support the manuscript for publication.

Reviewer #2 (Remarks to the Author):

Major concerns:

In the abstract, the authors wrote “.....play a leading role in oocyte aging in both mouse and human oocytes.”. But based on their current data, I can't find sufficient evidence to support such a strong claim, and it should be toned down.

The authors used the ERCC spike-in to normalize the amount of total RNA in the samples, and attributed the differences in the extent and number of down-regulated transcripts between this study and several previous studies to ERCC spike-in normalization. And then they claim that “our results updated the description of aging-associated transcriptome changes in GV oocytes and emphasized the need for proper normalization in analyzing dynamic transcriptome changes” (lines 446-448). To support this conclusion, at least the authors should analyze their data with or without the ERCC spike-in normalization and compare the results.

Minor concerns:

Fig.3C and Fig.6B, the left panels have been modified as explained in the rebuttal letter. But the right panels appear to have not been changed.

The words in Fig.s2A need to be aligned.

Line 297, “Compared to WT, 282 and 641 transcripts were increased or decreased more than 52-fold in Cxyc1 null oocytes at the GV stage, respectively (Fig. 6A)”. Please check whether the number of transcripts is correct.

Line 513, “Poly(A) tails (~200-444 250 bp) were added to transcribed mRNAs using the mMACHINE Kit (Invitrogen, AM1350)”, or it should be “Poly(A) tailing kit”?

Line 567, the authors described the lysis buffer containing 100 pg ERCC mRNA spike-in. The ERCC spike-in is usually used as xx μ l 1:xx dilution (ie. 0.1 μ l 1:80,000 dilutions of ERCC spike-in mix1 was used; see PMID: 30050112 or ERCC user guide).

Line 818 and line 821, “Scale bar = 10 μ M” should be “Scale bar = 10 μ m”.

Responses to Reviewers' Comments

Reviewer #2 (Remarks to the Author):

Major concerns:

In the abstract, the authors wrote “.....play a leading role in oocyte aging in both mouse and human oocytes.”. But based on their current data, I can't find sufficient evidence to support such a strong claim, and it should be toned down.

Response: Thank you very much for your suggestions. We have made the correction as suggested.

The authors used the ERCC spike-in to normalize the amount of total RNA in the samples, and attributed the differences in the extent and number of down-regulated transcripts between this study and several previous studies to ERCC spike-in normalization. And then they claim that “our results updated the description of aging-associated transcriptome changes in GV oocytes and emphasized the need for proper normalization in analyzing dynamic transcriptome changes” (lines 446-448). To support this conclusion, at least the authors should analyze their data with or without the ERCC spike-in normalization and compare the results.

Response: Thank you very much for your suggestions. We have analyzed our data without the ERCC spike-in normalization. The results showed that the number of differentially expressed genes significantly reduced compared to those with ERCC spike-in normalization. Moreover, without ERCC spike-in normalization, the pattern of aging-associated transcriptome changes was insignificant due to the absence of proper normalization. We have added this analysis result in the revised Supplementary Figure 1C and D.

Minor concerns:

Fig.3C and Fig.6B, the left panels have been modified as explained in the rebuttal letter. But the right panels appear to have not been changed.

Response: We are sorry for the mistake, the right panels have been modified along with the left panels, however the p-values were not updated. Now we have updated the right panels in Fig.3c and Fig.6b.

The words in Fig.s2A need to be aligned.

Response: We have made the correction as suggested.

Line 297, “Compared to WT, 282 and 641 transcripts were increased or decreased more than 52-fold in Cxhc1 null oocytes at the GV stage, respectively (Fig. 6A)”. Please check whether the number of transcripts is correct.

Response: We are sorry for the oversight. We have made the correction.

Line 513, “Poly(A) tails (~200-444 250 bp) were added to transcribed mRNAs using the mMACHINE Kit (Invitrogen, AM1350)”, or it should be “Poly(A) tailing kit”?

Response: We have made the correction as suggested.

Line 567, the authors described the lysis buffer containing 100 pg ERCC mRNA spike-in. The

ERCC spike-in is usually used as xx μl 1:xx dilution (ie. 0.1 μl 1:80,000 dilutions of ERCC spike-in mix1 was used; see PMID: 30050112 or ERCC user guide).

Response: We have re-described the use of ERCC spike-in in the methods. We used 0.5 μl 1:10000 dilutions of ERCC spike-in mix in human oocyte samples and 0.2 μl 1:1000 dilutions in mouse oocyte samples.

Line 818 and line 821, "Scale bar = 10 μM " should be "Scale bar = 10 μm ".

Response: We have made the correction as suggested.